# No Train No Gain: Revisiting Efficient Training Algorithms For Transformer-based Language Models

**Jean Kaddour**[1]*    **Oscar Key**[1]*    **Piotr Nawrot**[2]    **Pasquale Minervini**[2]    **Matt J. Kusner**[1]

[1]Centre for Artificial Intelligence, University College London
[2]School of Informatics, University of Edinburgh
{jean.kaddour.20, oscar.key.20, m.kusner}@ucl.ac.uk
{piotr.nawrot, p.minervini}@ed.ac.uk

## Abstract

The computation necessary for training Transformer-based language models has skyrocketed in recent years. This trend has motivated research on efficient training algorithms designed to improve training, validation, and downstream performance faster than standard training. In this work, we revisit three categories of such algorithms: **dynamic architectures** (layer stacking, layer dropping), **batch selection** (selective backprop, RHO loss), and **efficient optimizers** (Lion, Sophia). When pre-training BERT and T5 with a fixed computation budget using such methods, we find that their training, validation, and downstream gains vanish compared to a baseline with a fully-decayed learning rate. We define an evaluation protocol that enables computation to be done on arbitrary machines by mapping all computation time to a reference machine which we call *reference system time*. We discuss the limitations of our proposed protocol and release our code to encourage rigorous research in efficient training procedures: https://github.com/JeanKaddour/NoTrainNoGain.

## 1 Introduction

Language models are advancing rapidly, surpassing human-level performances in some experimental setups [11, 47, 99, 19, 12]. These improvements are primarily due to model, data, and training budget scaling [44, 34, 3]. Training a single state-of-the-art language model requires hundreds of thousands of GPU hours, costs millions of dollars, and consumes as much energy as multiple average US family households per year [81, 73, 14, 92, 43].

To remedy this, there is a rapidly growing line of work on *efficient training* algorithms, which modify the training procedure to save computation [81, 8, 112, 85, 64, 43]. Specifically, three distinct classes of such algorithms are **dynamic architectures** (layer stacking [30] and layer dropping [107]) which ignore some of the weights during training, **batch selection** (selective backprop [39] and RHO loss [66]) which skip irrelevant data, and **efficient optimizers** (Lion [13] and Sophia [58]) which claim to convergence faster than Adam(W) [48, 62].

However, we find that the evaluation methodology is not standardized, and there are inconsistencies in the quantification of a "speed-up". Two general trends are: comparisons of (1) **intermediate performances** throughout training instead of final ones within a pre-defined training budget, and (2) **incomplete speedup metrics**, e.g., training progress as a function of the number of iterations (or epochs) instead of wall-clock computation times *despite unequal per-iteration costs*.

As pointed out by Dehghani et al. [20], Dahl et al. [17], this can be unfair to baselines, which were not tuned for efficiency within the same compute budget.

---

*Equal contribution, alphabetical order.

| | GLUE | | | SuperGLUE | | |
|---|---|---|---|---|---|---|
| | 6h | 12h | 24h | 6h | 12h | 24h |
| **Baseline** | **77.1 ± 0.2** | 77.8 ± 0.2 | **78.3 ± 1.1** | 57.8 ± 0.9 | **58.5 ± 1.1** | 58.6 ± 1.2 |
| **Layer stacking** | **76.8 ± 0.8** | **78.4 ± 0.4** | **79.4 ± 0.2** | **58.6 ± 1.0** | 57.9 ± 0.7 | **58.7 ± 2.0** |
| **Layer dropping** | **76.8 ± 0.3** | 78.1 ± 0.2 | 78.6 ± 0.1 | **58.6 ± 0.8** | **58.5 ± 0.7** | 58.4 ± 0.8 |
| **Selective backprop** | 75.3 ± 0.6 | 76.6 ± 0.4 | 77.9 ± 0.3 | 57.4 ± 0.4 | **59.1 ± 0.9** | 58.5 ± 0.4 |
| **RHO loss** | 75.7 ± 0.1 | 76.5 ± 1.3 | 77.8 ± 0.2 | 57.6 ± 1.1 | 57.8 ± 0.6 | **58.7 ± 0.7** |
| **Baseline (BF16)** | **77.0 ± 0.3** | **77.8 ± 0.2** | **77.9 ± 0.3** | 57.6 ± 0.5 | 57.9 ± 0.5 | 57.9 ± 0.6 |
| **Lion (BF16)** | 62.0 ± 13.7 | 72.0 ± 0.5 | 71.4 ± 0.8 | 56.1 ± 2.5 | 57.5 ± 0.2 | 57.2 ± 2.3 |
| **Sophia (BF16)** | 73.9 ± 1.3 | 71.1 ± 4.2 | 72.3 ± 3.8 | **58.0 ± 0.6** | 57.8 ± 0.7 | 57.5 ± 0.7 |

Table 1: **Downstream performance, BERT.** Results for efficient training methods on the GLUE and SuperGLUE dev sets for three budgets (6 hours, 12 hours, and 24 hours) after pre-training a crammed BERT model [22, 29]. We report average validation of GLUE and SuperGLUE scores across all tasks (standard deviations for three seeds). We use mixed precision training with BF16 for the optimizer comparison as we found FP16 precision lead to numerical instabilities (Section 6).

An example for (1) includes learning rate schedules, which can be arbitrarily stretched to improve the final performance while "sacrificing" the quality of intermediate checkpoints [103, 49, 87]. This can make comparisons of intermediate performances unfair to baselines. For (2), additional regularization techniques can improve the per-iteration convergence at higher per-iteration costs [5, 27, 42, 17], rendering a wall-clock time comparison more appropriate.

| | SNI | | |
|---|---|---|---|
| | 6h | 12h | 24h |
| **Baseline** | 34.2 ± 0.2 | 38.3 ± 0.4 | **39.5 ± 0.1** |
| **Layer stacking** | **38.9 ± 0.2** | **39.2 ± 0.1** | 38.9 ± 0.2 |
| **Layer dropping** | 31.5 ± 0.3 | 34.4 ± 0.8 | 38.0 ± 0.5 |
| **Lion** | 20.8 ± 1.1 | 30.7 ± 0.4 | 33.7 ± 0.0 |
| **Sophia** | 26.7 ± 0.8 | 31.5 ± 0.8 | 34.1 ± 1.1 |

Table 2: **Downstream performance, T5-Base** [77, 69]. SNI [98] test ROUGE-L results (three seeds).

In this paper, we propose a simple evaluation protocol for comparing speedups of efficient training algorithms. We use this protocol to evaluate these algorithms for pre-training Transformer-based language models from scratch. We compare different training budgets (6, 12, and 24 hours), model architectures (BERT-Base [22] and T5-Base [77]), and (for batch selection algorithms) datasets (C4 [77], Wikipedia and BookCorpus [111], and MiniPile [41]). To account for variability in measured wall-clock time on different hardware and software configurations, we propose a simple measure that we call *reference system time* (RST).

Our key findings are as follows:

- **Training loss** (layer stacking, layer dropping, Lion, Sophia): The only approach to consistently outperform the training loss of the fully-decayed learning rate baseline across budgets and models is layer stacking (Lion matches this performance for certain BERT training budgets). This improvement reduces as the budget increases to 24 hours.

- **Validation loss** (selective backprop, RHO loss): Across three training datasets, none of the batch selection methods outperform the validation loss of the baseline.

- **Downstream tasks**: For a 24-hour budget, none of the efficient training algorithms we evaluate improves the downstream performance of the baseline.

- Methods with lower per-iteration costs than the baseline (i.e., **dynamic architecture** methods: layer stacking, layer dropping) can slightly improve downstream performance for lower budgets (6 hours, 12 hours), but the improvement disappears with longer training.

- Methods with higher per-iteration costs (i.e., **batch selection** methods: selective backprop, RHO loss, and some **efficient optimizer** methods: Sophia) are significantly worse than the baseline in some downstream tasks (GLUE, SNI), for all budgets.

- If we ignore the additional per-iteration computations of the three above methods, the downstream performance is still matched by the baseline.

## 2  Comparing Efficient Training Algorithms

What is the fairest way to compare efficient training algorithms? Ideally, each method should be given an identical compute budget, and methods that converge to better solutions than a baseline within this budget achieve a speed-up. Below, we discuss the issues of commonly used metrics to specify the compute budget, and propose an improved metric.

**The Pitfalls of Iterations**    Specifying a budget in number of training iterations is usually not suitable because the iteration time can vary between methods.

**The Pitfalls of FLOPs**    While FLOPs count the number of elementary arithmetic operations, they do not account for parallelizability (e.g., RNNs vs Transformers) or hardware-related details that can affect runtime. For example, FLOP counting ignores memory access times (e.g., due to different layouts of the data in memory) and communication overheads, among other things [20, 8].

**The Pitfalls of Wall-Clock Time**    WCT can fluctuate even on the same hardware, for instance, due to the usage of non-deterministic operations [2], hidden background processes, or inconsequential configurations, such as the clock rate. Further, we would like a metric that allows researchers to run on shared compute clusters, where hardware configurations will vary.

**Our Proposal: Reference System Time (RST)**    We propose a simple time measure that will allow us to standardize any timing result w.r.t. a reference hardware system (e.g., NVIDIA RTX 3090, CUDA runtime 11.8, PyTorch 2.0, etc.). We define the time elapsed on a reference system as *reference system time* (RST). To convert the training run of an arbitrary device to RST, we first record the time per training iteration on the reference training system.[3] Then, we compute the RST by multiplying the number of iterations run on the arbitrary device by the time per iteration on the reference system. This way, the time is grounded in practical time units but can be applied to any system.

### 2.1  The Pitfall of Comparing Intermediate Performances

One difficulty with comparing efficient training methods stems from the importance of the learning rate schedule to model performance. Various works have demonstrated that deep learning models tend to reach their maximum performance only once the learning rate has been fully decayed [109, 76]. If the learning rate schedule is stretched, this can delay the number of iterations required for the model to reach a given performance threshold since the optimizer keeps "exploring" the local loss basin, wasting iterations oscillating [33, 42].

This leads to unfair comparisons when the learning rate schedule is not set fairly between methods; a non-obvious pitfall that has been discussed previously in the literature [50, 20], but which we still observed in several of the works we evaluated [30, 107, 39, 58]. Consider comparing a baseline B with an efficient training method X, where both methods are trained for the same number of iterations, and the learning rate is decayed in terms of the number of iterations. B and X reach a similar performance at the end of training, but because X completes the iterations in less WCT, a speed-up is declared.

However, this is not a fair comparison because the training budget was measured in iterations, but the performance of the models was evaluated at a point specified in WCT, specifically the time it took X to complete the budget. Thus, B is evaluated at an intermediate point during training when its learning rate schedule has not been fully decayed. If the learning rate schedule of B had been shortened so it finished at the same WCT as X, the two methods may have reached similar performances in the same WCT time. The same problem arises when the budget is specified using different units to the point of measurement, including the cases where FLOPs are used for the budget.

To avoid this issue, each method should be trained to a fixed budget with the performance measured at the end of this budget. Importantly, the learning rate and other schedules must be decayed as a function of the same unit used to measure the budget.

---

[2] https://pytorch.org/docs/stable/notes/randomness.html
[3] We average the time required per iteration across 1000 iterations given some model architecture, batch, sequence length, system configuration, etc.

# 3 Experimental Setup

Given this computation measure, we can now evaluate efficient training algorithms under fixed computation budgets. We conduct experiments using two established and widely used Transformer language models: *BERT* [22] and *T5* [77]. We choose a single-GPU setting following recent works [29, 36, 69] to facilitate reproducibility and access for compute-restricted researchers. However, in principle, our protocol can be run on any hardware and straightforwardly used in a distributed setup.

**BERT: Encoder-only.** We follow the setup and hyperparameters of Geiping & Goldstein [29] with minor modifications. We pre-train a BERT-base-like model with 16 layers instead of 12, which consists of 120M parameters, using a masked language modeling (MLM) objective. We use the AdamW optimizer [62] with hyperparameters $\beta_1 = 0.9$, $\beta_2 = 0.98$, $\epsilon = 10^{-12}$, and weight decay of $10^{-2}$ [62]. Further, we use a one-cycle learning rate schedule [87] with a peak learning rate of $10^{-3}$ and gradient clipping of $0.5$. The batch size is $1536$ sequences, and the sequence length is $128$. We fine-tune and evaluate the pre-trained BERT models using the GLUE [94] and SuperGLUE [95] benchmarks. For fine-tuning on GLUE, we use the hyper-parameters from Geiping & Goldstein [29]. On SuperGLUE, we tune them using the validation accuracy of BoolQ [15]. We found inconsistencies in the literature on how to aggregate the SuperGLUE scores; here, we average the following scores: for CB, we use the F1 score; for the rest (BoolQ, COPA, RTE, WiC, WSC), we use the accuracy.

**T5: Encoder-Decoder.** We pre-train a T5v1.1-Base [77, 82] model using the original span-corrupting MLM objective and SentencePiece [52] tokenizer. We follow Nawrot [69] and use the AdamW optimizer [62] with tensor-wise LR scaling by its root mean square (RMS), base learning rate $0.02$, no weight decay, cosine schedule with final of $10^{-5}$ [63], gradient clipping of $1.0$, and $10^4$ warm up steps. We use a batch size of $144$ examples, where each example consists of an input of $512$ tokens and an output of $114$ tokens. We evaluate our pre-trained T5 models on the Super-Natural-Instructions [SNI, 98] benchmark. To fine-tune the model on SNI, we strictly

---

**Algorithm 1** Layer stacking [30]

---

**Input:** number of layers $L$, number of stacking operations $k$
Initialize model $f'_0$ with $L/2^k$ layers
$f_0 \leftarrow \text{Train}(f'_0)$
**for** $i \leftarrow 0, \ldots, k-1$ **do**
$\quad f'_i \leftarrow (f_i, f_i)$ {Stack layers.}
$\quad f_i \leftarrow \text{Train}(f'_i)$

---

follow the hyperparameters of [69, 98]. For both pre-training and fine-tuning, we use TF32 precision.

**Dataset.** Unless specified otherwise, we use the C4 [77] dataset for less than one epoch (i.e., without data repetitions) without sentence curriculum and de-duplication [53, 29]. In Section 5, we additionally use two other datasets.

**Learning-rate schedule.** We adjust the learning rate schedule based on the elapsed time conditional on a time budget (measured in RST), similar to [37, 29], who measure raw WCT.

**Hyper-parameter search.** For all considered methods, we tune their hyper-parameters based on the pre-training loss. We list all details about each method's hyper-parameters, our considered grid search ranges, and the best values we found in Appendix C.

**Reference System.** We record the RST on two separate systems for BERT and T5. For BERT, we choose a single NVIDIA RTX 3090, CUDA 11.7, PyTorch 1.13. For T5, we choose an NVIDIA A100, CUDA runtime 11.8, PyTorch 2.0.

# 4 Case Study 1: Dynamic Architectures

## 4.1 Layer stacking

*Layer stacking* [30], as summarized in Algorithm 1, replicates a $L$-layer model into a $2L$-layer model by copying its parameters, effectively warm-starting the stacked model with parameters transferred from the smaller model. Thereby, it benefits from faster per-iteration times in the early training phases when using fewer layers. Gong et al. [30] attribute the success of this method to attention

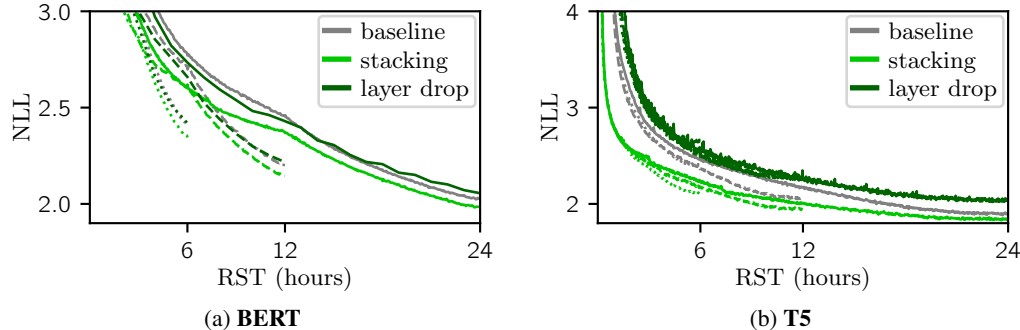

(a) **BERT**  (b) **T5**

Figure 1: **Training losses, dynamic architecture methods** (layer stacking, and layer dropping). Results are shown for RST budgets of 6 hours (······), 12 hours (- - -), and 24 hours (——), on C4.

distributions of bottom layers being very similar to attention distributions of top layers, indicating that their functionalities are similar.

## 4.2 Layer dropping

Layer dropping exploits the following observation: the layers of a network do not contribute equally to the loss reduction throughout training [35, 6, 106]. It does so by randomly choosing parts of the model to be skipped during each training step. Specifically, it replaces a subset of Transformer blocks with the identity function. As a result, it reduces the overall computational cost of the model for each training step because it skips these layers during the forward and backward passes.

To minimize the impact of layer dropping on the pretraining loss of the model, Zhang & He [107] employ a time and depth schedule that determines the probability of dropping each block. The time schedule begins with a zero probability of dropping each block. Then it increases this probability throughout the training process until it reaches a maximum of $(1-\bar{\alpha})$, where the hyperparameter $\bar{\alpha} = 0.5$ as chosen by Zhang & He [107].

The depth schedule ensures that blocks located earlier in the model are dropped with a lower probability than those located later/deeper. An important hyperparameter of the depth schedule in layer dropping is $\gamma_f$, which controls the rate at which the probability of dropping layers increases. A higher value of $\gamma_f$ leads to a quicker increase in the probability. Zhang & He [107] set $\gamma_f$ to 100 in their experiments. We refer the reader to the original work by Zhang & He [107] for more details.

---

**Algorithm 2** Layer dropping [107]

1: **Input:** iterations $T$, layer keep probability $\bar{\alpha}$, temperature budget $\gamma_f > 0$, layers $L$, functions (self-attention, layer-norm, feed-forward) $f_{ATTN}, f_{LN}, f_{FFN}$, loss function $\mathcal{L}$, data $(\mathbf{x}_0, \mathbf{y})$, output layer $f_O$
2: $\gamma \leftarrow \frac{\gamma_f}{T}$
3: **for** $t \leftarrow 1$ to $T$ **do**
4: $\quad p \leftarrow 1$ {Keep probability.}
5: $\quad \alpha_t \leftarrow (1-\bar{\alpha})\exp(-\gamma \cdot t) + \bar{\alpha}$
6: $\quad p_d \leftarrow \frac{1-\alpha_t}{L}$ {Layer decay.}
7: $\quad$ **for** $i \leftarrow 0$ to $L-1$ **do**
8: $\quad\quad s \sim \text{Bernoulli}(p)$ {Keep or drop.}
9: $\quad\quad$ **if** $s == 0$ **then**
10: $\quad\quad\quad \mathbf{x}_{i+1} \leftarrow \mathbf{x}_i$ {Drop.}
11: $\quad\quad$ **else**
12: $\quad\quad\quad \mathbf{x}'_i \leftarrow \mathbf{x}_i + \frac{f_{ATTN}(f_{LN}(\mathbf{x}_i))}{p}$
13: $\quad\quad\quad \mathbf{x}_{i+1} \leftarrow \mathbf{x}'_i + \frac{f_{FFN}(f_{LN}(\mathbf{x}'_i))}{p}$
14: $\quad\quad p \leftarrow p - p_d$ {Decay prob.}
15: $\quad \ell \leftarrow \mathcal{L}(f_O(\mathbf{x}_L), \mathbf{y})$
16: $\quad f_{ATTN}, f_{LN}, f_{FFN}, f_O \leftarrow \text{Update}(\ell)$

---

## 4.3 Results

**Training losses.** Figure 1 shows the pre-training losses for the baseline, layer stacking, and layer dropping on BERT and T5 when given a budget of 24 hours in RST. In both settings, layer stacking achieves the lowest loss within this budget. The gap between layer stacking and the baseline closes almost completely as the budget is increased to 24 hours. However, layer dropping is consistently worse than the baseline throughout training. In both models, the gap between layer dropping and the baseline is larger than between the baseline and layer stacking at the end of training. Further, layer dropping introduces additional fluctuations in the loss when training T5, compared to the baseline.

**Downstream performance.** Figure 2 shows the SuperGLUE downstream performance after using the baseline and all dynamic architecture methods (layer stacking and layer dropping) to pre-train

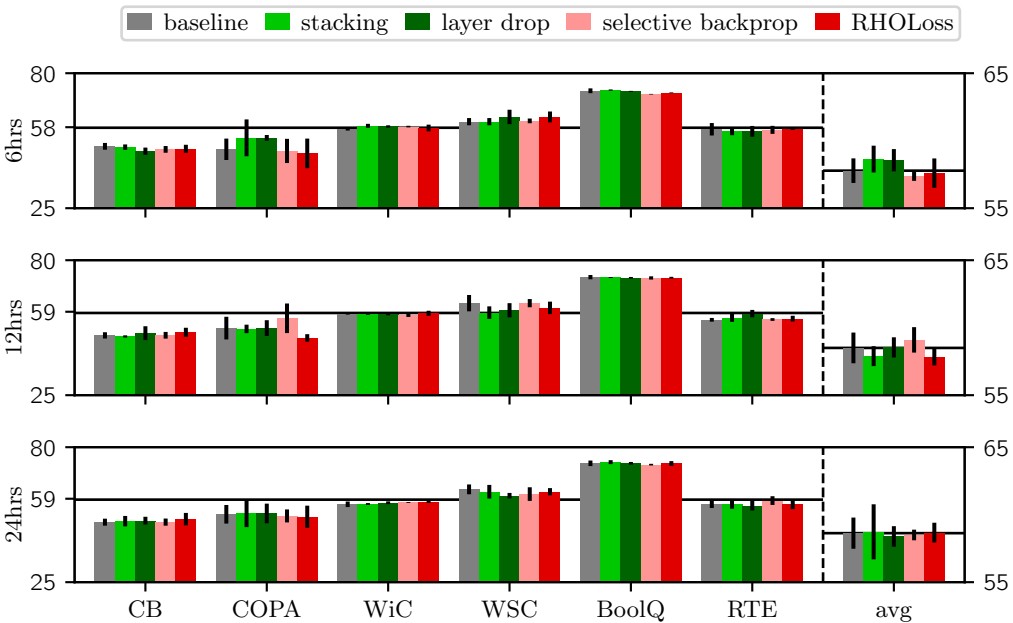

Figure 2: **BERT models evaluated on SuperGLUE.** The black vertical error bars indicate the standard deviation over three seeds. The black horizontal line shows the baseline average performance. For clarity, the individual tasks are plotted against the left-hand axis, while the average accuracy is plotted against the right-hand axis.

BERT. We evaluate all methods using three different RST budgets (to get a sense of the Pareto-frontier of budget and performance [75]): 6 hours, 12 hours, and 24 hours. We notice that on specific SuperGLUE datasets, the efficiency methods have little or even detrimental effect on performance (CB, WiC, WSC, BoolQ). Across all datasets, COPA appears to have the largest gains from efficient training. Averaged across all datasets, efficient training methods have little effect. We report the exact numbers in Table 1 (right). On average, across all budgets, neither layer stacking and layer dropping significantly improve over the baseline.

We also compare the performance of dynamic architecture methods methods on the T5-base model evaluated on the SNI benchmark and report the results in Table 2. Here layer stacking significantly improves over the baseline for 6 and 12 hour budgets. However, given 24 hours for training, the final accuracy of layer stacking reduces back to the accuracy of the 6 hour model. Here the baseline significantly improves over all methods. Notably, for all budgets, layer dropping is consistently outperformed by the baseline.

## 5 Case Study 2: Batch Selection

Scaling up the size of web-crawled data for pre-training has been one of the major drivers to improve the capabilities of language models [44, 10]. A line of work has argued that training speed-ups emerge through the selection of *informative* examples. They argue that these examples can be identified from certain statistics collected throughout training [4, 45, 39, 46, 72]. Here, we focus on two such methods that directly alter the training procedure to steer gradient computations towards informative samples by subsampling examples from a *mega-batch* to curate the mini-batch, called *batch selection*.

### 5.1 Selective Backprop

Due to its simplicity, we first examine selective backprop [39], described in Algorithm 3. The high-level idea of selective backprop is to compute the backward pass only on the training examples with the highest loss. To construct such batches, it first computes the loss of every example in a

**Algorithm 3** Selective backprop [39]

1: **Input:** iterations $T$, data $\{(\mathbf{x}_i, \mathbf{y}_i)\}_{i=1}^N$, loss $\mathcal{L}$, mini-batch size $B_m$, mega-batch size $B_M$, number of inputs to compute loss CDF $R$, selectivity $\beta > 0$, model $\boldsymbol{\theta}$, loss history $\mathcal{H} = ()$
2: **for** $t \leftarrow 1$ to $T$ **do**
3:    $\mathcal{B}_m \leftarrow \{\}$ {Initialize mini-batch.}
4:    $\mathcal{B}_M \subset \{(\mathbf{x}_i, \mathbf{y}_i)\}_{i=1}^N$ {Select mega-batch.}
5:    $\{\ell_j\}_{j=1}^{B_M} \leftarrow \mathcal{L}(\mathcal{B}_M; \boldsymbol{\theta})$ {Compute loss.}
6:    **for** $j \leftarrow 1$ to $B_M$ **do**
7:       $\mathcal{H} \leftarrow (\ell_j, \mathcal{H})$ {Add to history}
8:       $p \leftarrow \text{CDF}(\ell_j; \mathcal{H}_{1,\ldots,R})^\beta$ {Selection prob.}
9:       $s \sim \text{Bernoulli}(p)$ {Select or not.}
10:      **if** $s == 1$ **then**
11:        $\mathcal{B}_m \leftarrow \mathcal{B}_m \cup (\mathbf{x}_j, \mathbf{y}_j)$
12:      **if** $|\mathcal{B}_m| == B_m$ **then**
13:        $\boldsymbol{\theta} \leftarrow \text{Update}(\boldsymbol{\theta}, \mathcal{B}_m)$ {Backwards pass.}
14:        $\mathcal{B}_m \leftarrow \{\}$

**Algorithm 4** RHO loss [66]

1: **Input:** iterations $T$, train data $\mathcal{D}_{\text{train}}$, validation data $\mathcal{D}_{\text{val}}$, loss $\mathcal{L}$, small model $\boldsymbol{\theta}^S$ trained on $\mathcal{D}_{\text{val}}$, mini-batch size $B_m$, mega-batch size $B_M$, learning rate $\eta$, model $\boldsymbol{\theta}$
2: **for** $(x_i, y_i) \in \mathcal{D}_{\text{train}}$ **do**
3:    $\ell_i^S \leftarrow \mathcal{L}(x_i, y_i; \boldsymbol{\theta}^S)$ {Small model loss on train}
4: **for** $t \leftarrow 1$ to $T$ **do**
5:    $\mathcal{B}_M \subset \{(\mathbf{x}_i, \mathbf{y}_i)\}_{i=1}^N$ {Select mega-batch.}
6:    $\{\ell_j\}_{j=1}^{B_M} \leftarrow \mathcal{L}(\mathcal{B}_M; \boldsymbol{\theta})$ {Compute loss.}
7:    $\mathcal{R} \leftarrow \{\ell_j - \ell_j^S \mid j \in \mathcal{B}_M\}$
8:    $\mathcal{R}_{\text{top}} \leftarrow \max_{B_m}(\mathcal{R})$ {top $B_m$ of $\mathcal{R}$}
9:    $\boldsymbol{\theta} \leftarrow \text{Update}(\boldsymbol{\theta}, \mathcal{R}_{\text{top}})$ {Backwards pass.}

| | 6h | 12h | 24h |
|---|---|---|---|
| Baseline | **2.42 ± 0.00** | **2.20 ± 0.00** | **2.10 ± 0.01** |
| Selective backprop | 2.73 ± 0.18 | 2.44 ± 0.06 | 2.18 ± 0.02 |
| RHO loss | 2.61 ± 0.00 | 2.37 ± 0.00 | 2.16 ± 0.01 |

Table 3: **Validation losses, batch selection methods** (selective backprop, and RHO loss). Results are shown for RST budgets of 6, 12, and 24 hours, on C4.

| | Val. Loss | GLUE |
|---|---|---|
| Baseline | 2.21 | **77.79 ± 0.2** |
| Selective backprop | 2.23 | **77.92 ± 0.1** |
| RHO loss | **2.19** | 77.16 ± 0.4 |

Table 4: **Batch Selection For Free.** Results for a 12-hour RST budget on C4, removing all costs used to select batches.

uniformly-sampled mega-batch. It then samples a high-loss subset of the mega-batch by ranking points based on their loss percentiles w.r.t. historical losses among recently-seen inputs.

## 5.2 RHO Loss

Mindermann et al. [66] argue that solely prioritizing high training loss results in two types of examples that are unwanted: (i) mislabeled and ambiguous data, as commonly found in noisy, web-crawled data; and (ii) outliers, which are less likely to appear at test time. The authors propose down-weighting such data via a selection objective called *Reducible Holdout (RHO) loss*, shown in Algorithm 4.

The authors acknowledge that their method comes with three overhead costs: (1) pre-training a proxy model using holdout data, (2) one extra forward pass over the entire training set to store the proxy model's training losses, and (3) additional forward passes for the batch selection. In our evaluations we never count the costs of (1) and (2) because, as Mindermann et al. [66] point out, these costs can be amortized over many training runs. For (1), we pre-train a model for 6 hours of RST on held-out data, which is enough to reach reasonable performances (Table 1).

Despite Mindermann et al. [66] motivating their method for a scenario where additional workers are available to perform batch selection, we wondered if it might still provide performance gains even if this is not the case. Mindermann et al. [66] do not implement or evaluate an algorithm where extra workers are used to select batches. Because of this, we evaluate RHO loss under two protocols: in the main results, we count the batch selection costs (3) against the training budget, while in Section 5.4, we provide a second set of results where we ignore the cost of batch selection.

## 5.3 Results

We assume that the effects of selecting better training batches should be largely agnostic to whether we pre-train a BERT or T5 model. Hence, instead of training both architectures, we decide to pre-train only BERT models and instead vary the datasets and budgets as follows.

For the first set of experiments, we fix the budget to 12 hours and consider three different pre-training datasets: (i) C4 [77], consisting only of webpage text which, despite being regularly used for pre-

**Algorithm 5** Sophia [58]

1: **Input:** $\boldsymbol{\theta}_1$, learning rate $\{\eta_t\}_{t=1}^T$, hyper-parameters $\{\lambda, \beta_1, \beta_2, \epsilon, \rho, k\}$, and GNB-Estimator
2: Set $m_0 = 0, v_0 = 0, h_{1-k} = 0$
3: **for** $t = 1$ **to** $T$ **do**
4:      Compute minibach loss $\mathcal{L}_t(\boldsymbol{\theta}_t)$.
5:      Compute $g_t = \nabla\mathcal{L}_t(\boldsymbol{\theta}_t)$.
6:      $m_t = \beta_1 m_{t-1} + (1 - \beta_1)g_t$
7:      **if** $t \bmod k = 1$ **then**
8:          Compute $\hat{h}_t = \text{GNB}(\boldsymbol{\theta}_t)$.
9:          $h_t = \beta_2 h_{t-k} + (1 - \beta_2)\hat{h}_t$
10:      **else**
11:          $h_t = h_{t-1}$
12:      $\boldsymbol{\theta}_t = \boldsymbol{\theta}_t - \eta_t\lambda\boldsymbol{\theta}_t$ (weight decay)
13:      $\boldsymbol{\theta}_{t+1} = \boldsymbol{\theta}_t - \eta_t \cdot \text{clip}(m_t/\max\{h_t, \epsilon\}, \rho)$

**Algorithm 6** Lion [13]

1: **Input:** $\boldsymbol{\theta}_1$, learning rate $\{\eta_t\}_{t=1}^T$, hyperparameters $\{\lambda, \beta_1, \beta_2\}$
2: Set $m_0 = 0$
3: **for** $t = 1$ **to** $T$ **do**
4:      Compute minibach loss $\mathcal{L}_t(\boldsymbol{\theta}_t)$.
5:      Compute $g_t = \nabla\mathcal{L}_t(\boldsymbol{\theta}_t)$.
6:      $u_t = \text{sign}\left(\beta_1 m_{t-1} + (1 - \beta_1)g_t\right)$
7:      $m_t = \beta_2 m_{t-1} + (1 - \beta_2)g_t$
8:      $\boldsymbol{\theta}_{t+1} = \boldsymbol{\theta}_t - \eta_t u_t$

training, is known to have quality issues [53], (ii) Bookcorpus and Wikipedia [22], which contain polished, book(-like) text and MiniPile [41], a subset of the diverse Pile pre-training corpus [28], containing code, mathematics, books, webpages, and other scientific articles.

**Validation loss.** To rank the pre-training performances, we compare the **validation loss**, which is directly comparable as we use the same inputs for all methods (which is not the case for the training data). This is shown in Figure 3, which shows the validation loss every 3 hours throughout training. We find that both batch selection methods do not improve over the baseline.

**Downstream performance.** Next, we investigate downstream performances, fix the C4 corpus as the pre-training corpus, and vary the budgets (6, 12, and 24 hours). Figures 2 and 16 show the results on SuperGLUE and GLUE. We observe very small differences between the methods and the baseline. On average (Table 1) no batch selection method significantly outperforms the baseline.

### 5.4 Ablation: Batch Selection For Free

We want to disentangle whether batch selection methods fail to improve over the baseline because their gains do not compensate for their computational overhead. In the previous section (Section 5.3) and the main results (Table 1),

Figure 3: **Validation losses for different datasets.** Results for batch selection methods (selective backprop and RHO loss) for a 12-hour RST budget.

we accounted for this overhead. Therefore, in those experiments, selective backprop and RHO loss effectively update the model for fewer iterations than the baseline within the fixed budgets. Here, we re-run a small number of experiments where batch selection is free, and so we train these models with the same number of iterations as the baseline.

Specifically, we run an experiment for 12 hours using the C4 corpus and GLUE downstream tasks. For selective backprop, we choose $\beta = 1$, which resulted in ~1.7x of the wall-clock time; while for RHO loss, we choose a mega-batch size that is 10x larger (15360) than the mini-batch size (1536), following Mindermann et al. [66], which led to ~5.3x of the original pre-training time. Table 4 shows the results. We see that RHO loss reaches a slightly better final validation loss but performs worse on the GLUE downstream tasks than baseline and selective backprop, which does not improve over the baseline.

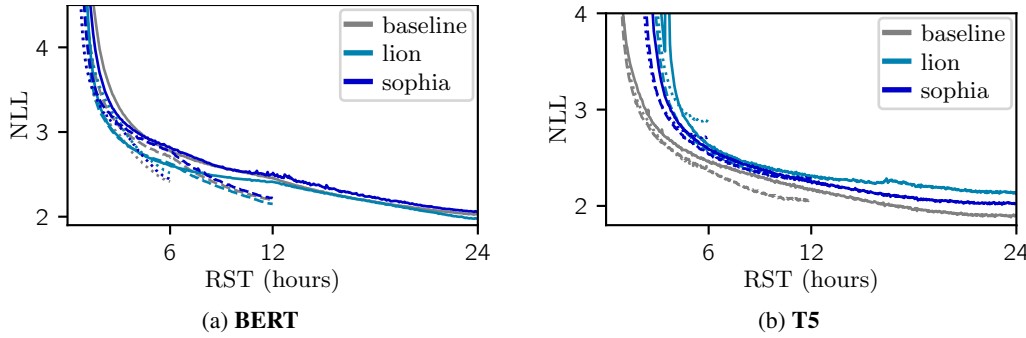

(a) **BERT**    (b) **T5**

Figure 5: **Training losses, efficient optimizer methods** (Lion, and Sophia). Results are shown for RST budgets of 6 hours (⋯⋯), 12 hours (- - -), and 24 hours (——), on C4.

# 6 Case Study 3: Efficient Optimizers

Recently, two efficient optimizers were proposed to replace the ubiquitous Adam(W) optimizers [48, 62] for language models: Lion [13] and Sophia [58].

Lion [13] is an optimizer symbolically discovered in the vast program space of first-order optimization primitives. As such, it does not follow any theory-grounded principle that would justify its acceleration property a priori; however, Chen et al. [13] report empirical speed-ups over AdamW in many settings.

Sophia [58] is a scalable stochastic second-order optimizer primarily designed for and evaluated on language model pre-training. The authors claim that Sophia achieves a 2x speed-up compared with AdamW in the number of steps, total compute, and wall-clock time. The authors study two Hessian estimators, but as of this writing, only open-source the code for the empirically better one (Gauss-Newton-Bartlett), which we use here.

The baseline in this section simply refers to AdamW [62].

**Mixed Precision Training with BF16 vs. FP16**  A common practice for training language models is to use mixed precision training [65]. In initial experiments with BERT, we observed several numerical instabilities (NaN training losses) during hyper-parameter search after inserting Lion and Sophia into our training pipelines as drop-in replacements. Our default mixed-precision mode was FP16, and as noted by other practitioners of Sophia [57], BF16 can sometimes be more stable. Hence; for the optimizer comparisons with BERT, we report results in BF16; including the baseline (although we notice that the baseline's training curves are essentially identical across both modes). For the T5 model, we use the TF32 precision format.

**RMS-Scaling for T5 Pre-training**  T5 pre-training [77, 82] typically employs the Adafactor optimizer [83], which relies on tensorwise learning rate scaling determined by the tensor's root mean square (RMS). This scaling has been identified as critical for convergence during pre-training when using AdamW [69]. Initial tests with Lion and Sophia without additional adjustments led to divergence for higher learning rates or suboptimal performance. This mirrored the behavior of AdamW without RMS scaling. To address this, we incorporated RMS scaling into Lion and Sophia for a subsequent round of experiments, which we outline in Appendix C.6.

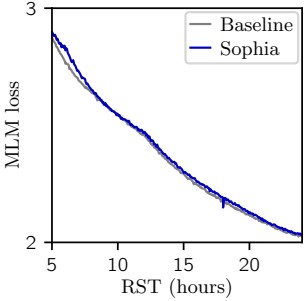

## 6.1 Results

In the case of BERT downstream performances (Table 1), we find that Lion and Sophia perform about the same as the baseline. Figure 6 shows the results in more detail. We note that baseline (AdamW) consistently ranks the highest and has a comparatively low standard deviation over random seeds.

Figure 4: BERT Validation loss when we do not count Sophia's Hessian update steps.

Analogous to the batch selection for free ablation in Section 5.4, we also experiment with Sophia while not counting for Hessian update steps and running for the same number of iterations, as shown in Figure 4. Surprisingly, we still do not observe any speedup.

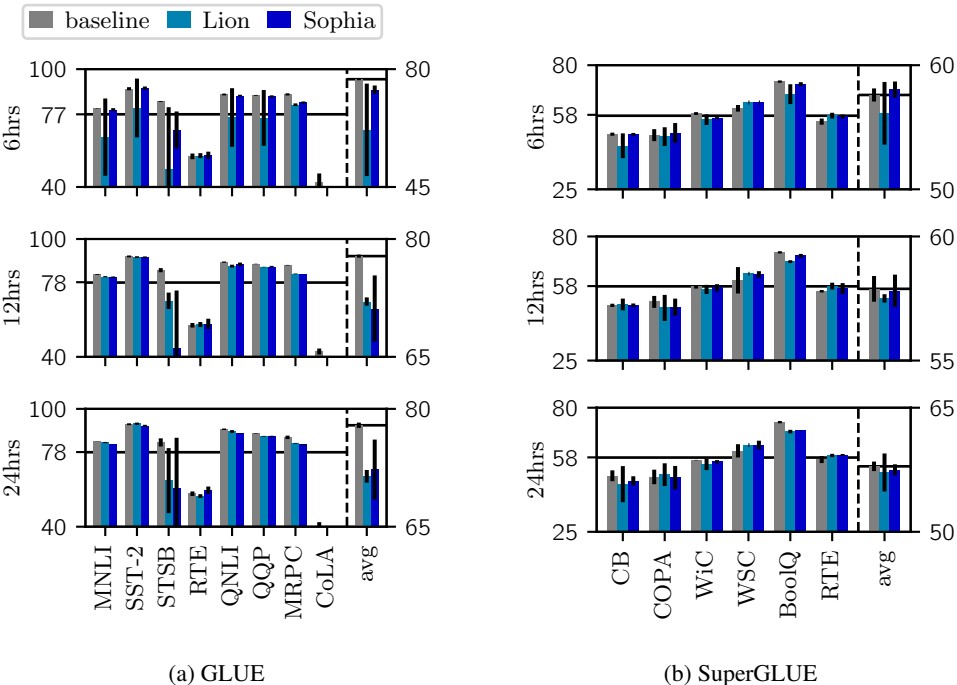

(a) GLUE

(b) SuperGLUE

Figure 6: **BERT BF16 models evaluated on (Super-)GLUE.** The black vertical error bars indicate the standard deviation over three seeds. The black horizontal line shows the average performance of the baseline. For clarity, the individual tasks are plotted against the left-hand axis, while the average accuracy is plotted against the right-hand axis.

## 7   Conclusion

In this work, we closely examined efficient training algorithms which promised to deliver training speed-ups. First, we clarify that quantifying a training speed-up without specifying an explicit training budget can be misleading and that some previous works missed this. To normalize the wall clock time across different hardware, we introduce the reference system time measure. Then, we put three classes of algorithms to the test in various pre-training settings of BERT and T5 models. We found only a few settings where some of the considered algorithms improved over the baseline.

## Acknowledgments

The authors would like to thank Edoardo Ponti for his feedback on an earlier version of this manuscript. JK and OK acknowledge support from the Engineering and Physical Sciences Research Council with grant number EP/S021566/1. OK was supported by G-Research. PN was supported by the UKRI Centre for Doctoral Training in Natural Language Processing, funded by the UKRI (grant EP/S022481/1) and the University of Edinburgh, School of Informatics and School of Philosophy, Psychology & Language Sciences. PM was partially funded by the European Union's Horizon 2020 research and innovation programme under grant agreement no. 875160, ELIAI (The Edinburgh Laboratory for Integrated Artificial Intelligence) EPSRC (grant no. EP/W002876/1), an industry grant from Cisco, and a donation from Accenture LLP; and is grateful to NVIDIA for the GPU donations. This work was supported by the Edinburgh International Data Facility (EIDF) and the Data-Driven Innovation Programme at the University of Edinburgh.

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

# A    Limitations and Future Work

Firstly, while we ran extensive ablations within the 6-, 12-, and 24-hour training regimes, it is possible that our results do not generalize to much longer ones. We justify this choice for two reasons. Firstly, all downstream task performances observed are only a few percentage points worse than state-of-the-art performances; e.g., our 24-hour BERT model reaches ~79% / 58.5% on GLUE/SuperGLUE, while fine-tuning the original BERT-base model[4] reaches 80.9% / 60.8%, respectively. Similarly, our 24h-T5 model reaches 39.5% on SNI, while using the original checkpoint, reaches 41.0% [69]. Secondly, we argue that an efficient training method should work well specifically in settings with limited budgets.

Next, as illustrated in Appendix B, there is an abundance of efficient training algorithms, and rigorously evaluating all of them is prohibitively expensive. Hence, one limitation of this work remains that we did not consider other efficiency-promoting algorithms, and we hope to explore more in future work.

Another limitation is our sole focus on language model pre-training. Investigating approaches for (i) efficient fine-tuning of (large) language models or (ii) pre-training on other data-intensive modalities such as images and video remains promising too. We expect that our experimental protocol utilizing RSTs will benefit future works doing so.

# B    Related Work

## B.1    Efficient Training Algorithms

There is an abundance of proposals for efficient training; surveys of these can be found in [8, 85, 43]. We roughly categorize them into *architecture-centric*, *data-centric*, *optimization-centric*, and *others*.

**Architecture-centric strategies.**    These decide how to avoid forward/backward passes of specific weights in the network. The idea of gradually growing a network to accelerate training, as in layer stacking, dates back to the 90s [25, 55]. There are a number of follow-ups to the layer stacking paper [30], including automated stacking using elastic supernets [56] loss- and training-dynamics preserved stacking [86] and stacking small pre-trained models to train larger models [96].

Methods akin to layer dropping include FreezeOut [9], LayerDrop[26] (which focuses on network sparsity), and AutoFreeze [60]. In these methods, forward/backward passes for specific layers are skipped based on either a pre-determined schedule [9, 26] or based on statistics from prior forward/backward passes [60].

While not the focus of this work, a similar motivation can also be found in dynamic sparse training methods, which aim to identify relevant sub-networks during training and promote the prunability of the network [67, 21, 24]. However, these approaches do typically not lead to training speedups since unstructured sparsity is not GPU-friendly, which is why we do not consider them here [59].

**Data-centric strategies.**    Besides batch selection methods, which subsample relevant data from a mega-batch throughout training as discussed in Section 5, there are a number of other approaches aiming to either order or subsample training data. We did not consider them in our experiments since they either have a different motivation than training efficiency or are not directly suitable for a drop-in modification of the training procedure. However, we briefly summarize three classes of such work. Firstly, one of the oldest classes of data-selection strategies for training is *curriculum learning* (for a recent survey, see Wang et al. [97]). We do not consider curriculum learning here as it has already been extensively evaluated [100], and because it was initially motivated to improve generalization rather than achieve efficiency gains. Secondly, *task-specific retrieval* [32, 105] use task-specific data to retrieve similar data from a large unsupervised corpus. Different from the task-agnostic batch selection methods we consider here, these methods are specifically designed to improve downstream task performance. Lastly, another line of work tries to subsample the entire dataset decoupled from the training procedure by various scoring heuristics [16, 31, 74, 54, 88, 1, 102, 101, 61], and we believe that further investigating such can be a promising direction for future work.

---

[4] https://huggingface.co/bert-base-uncased

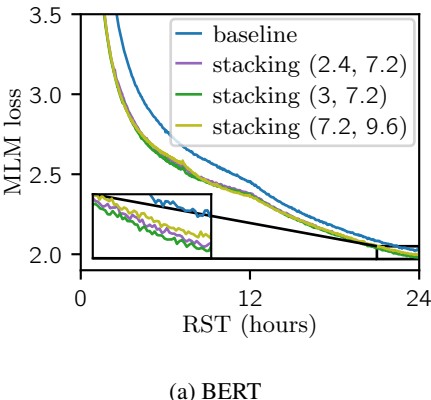
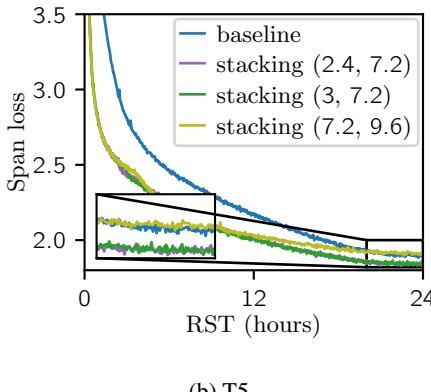

|              |              |
|:------------:|:------------:|
| (a) BERT     | (b) T5       |

Figure 7: Layer stacking grid search: we tune the intervals at which the model is doubled. The notation "stacking $(a, b)$" signifies that the model's size was doubled once at $a$ hours and then again at $b$ hours, measured using RST.

**Optimization-centric strategies.** Lots of optimizers have been proposed with the goal of speeding up convergence [23, 108, 113]; yet, Schmidt et al. [80] report that none of them consistently outperforms Adam(W) [48, 62] when put to a rigorous test. Instead of modifying the training procedure, another line of work observed intermediate performance speedups by averaging weights along the training trajectory post-hoc training [40]; however, such gains arise primarily through effectively lowering the learning rates without directly intervening in the training process [78, 79], which is different to the in this work considered methods which do intervene.

**Others.** These include ways to improve the faster computation of Transformer building blocks [91, 18, 70], allocate compute conditioned on the input [84, 71, 2], network initialization [7, 110].

## B.2 Efficient Training Meta-Studies

**Budget-centric recipes.** A different line of work investigates budget-centric training recipes, for example, for academic settings with multiple GPUs [38, 114], or hard time constraints such as training on a single GPU for one day [36, 29, 69]. Our work adopts some of the recipes proposed by Geiping & Goldstein [29], Nawrot [69] and aims to complement them by investigate (other) speedup techniques.

**Empirical Meta-Studies of Training Transformer-based models.** Narang et al. [68] study Transformer modifications across implementations and applications, finding that most do not meaningfully improve performances. Similarly, Tay et al. [90] study the scaling properties of various Transformer-based architectures (some of which are designed for efficiency), concluding that the original Transformer proposed by Vaswani et al. [93] has the best scaling behavior. Dehghani et al. [20] discuss how common cost indicators of machine learning models have different pros and cons and how they can contradict each other. Further, they show how training time can be gamed. Our work is heavily inspired by Dehghani et al. [20] and aims to complement it in two ways: (1) by proposing to normalize WCT across different systems (using RST) and (2) by a thorough experimental study in the case of pre-training Transformer-based language models.

Closest to our work is the concurrent work by Dahl et al. [17], who exhaustively discuss various pitfalls of benchmarking neural network optimizers. Among other considerations, they propose to benchmark algorithms within a fixed runtime budget, a practice we agree with and use in our work too. We believe our work additionally complements theirs by other methods beyond optimizers.

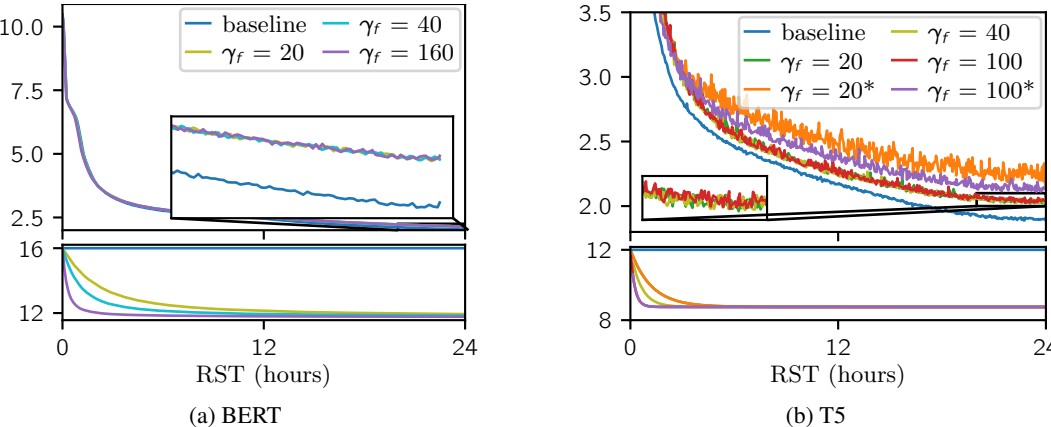

(a) BERT

(b) T5

Figure 8: **Grid search** performed on the hyperparameter $\gamma_f$ for layer dropping in a 24-hour budget setting. For the T5 model with layer dropping we also test smaller learning rate $(1e-2)$, because we observe instabilities with the original one $(2e-2)$. We mark runs with learning rate $= 2e-2$ with a $*$. The upper plots depict the training loss for each method, while the lower plots showcase the average number of active layers.

## C Hyper-Parameter Search

### C.1 Layer stacking: When To Stack

Figure 7 illustrates that layer stacking has relatively low sensitivity to different stacking RST hour times, namely $\{(2.4, 7.2), (3, 7.2), (7.2, 9.6)\}$, with $\{(2.4, 7.2), (3, 7.2)\}$ yielding similar performance and $(7.2, 9.6)$ slightly underperforming. For our experiments in Section 4.3, for both BERT and T5, we choose $(3, 7.2)$, as it maintains the same $\frac{\text{stacking step}}{\text{all training steps}}$ ratio proposed by Gong et al. [30].

### C.2 Layer dropping: How to Drop

In Figure 8, we observe that different choices of $\gamma_f$ yield comparable performance for both the BERT and T5 models. However, the layer dropping training curves for the T5 model are notably spikier than those of the baseline, suggesting that the layer dropping method demonstrates less stability during training with this architecture. As a result, we also tested a smaller learning rate $(1e-2)$, in contrast to the original training rate $(2e-2)$, which ultimately yielded better results. The parameters selected for our experiments in Section 4.3 were $\gamma_f = 20, lr = 1e-2$ for T5 and $\gamma_f = 100$ for BERT.

### C.3 Selective backprop: Selectivity Scale

We tune the selectivity scale $\beta$, where $\beta \in \{1, 2, 3\}$. Jiang et al. [39] use 33% and 50% selectivity in their experiments, which approximately corresponds to $\beta = \{1, 2\}$, respectively. We find that the larger the $\beta$ value, the worse the pre-training performance. Note that the higher the $\beta$ value, the more forward passes selective backprop needs to perform in order to collect enough samples for a backward pass, which decreases the total number of parameter update steps within the RST budget. For the experiments in Section 5.3, we chose $\beta = 1$, as it consistently achieves the best performance.

### C.4 RHO loss: Mega-batch Size

RHO loss requires one additional hyper-parameter, the size of the mega-batch, from which the mini-batch then gets subsampled from. We tune this hyper-parameter in Figure 10a and similar to Appendix C.3, we find that the larger this size gets, the worse the validation loss. We started tuning it based on BCWK, and given the clear hierarchy we observed; we decided not to tune it on other datasets and simply set it to 2x (3072).

Another implicit set of hyper-parameters is how to pre-train the proxy/irreducible loss models. Here, we follow suggestions by Mindermann et al. [66] and choose the same architecture as the target model.

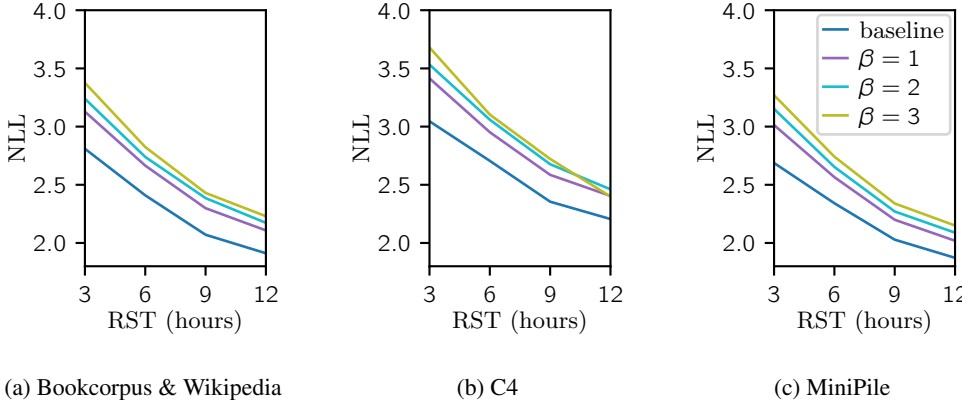

| (a) Bookcorpus & Wikipedia | (b) C4 | (c) MiniPile |

Figure 9: Selective backprop grid search: we tune the $\beta$ hyperparameter. Each plot shows the validation loss over time during training for the given dataset.

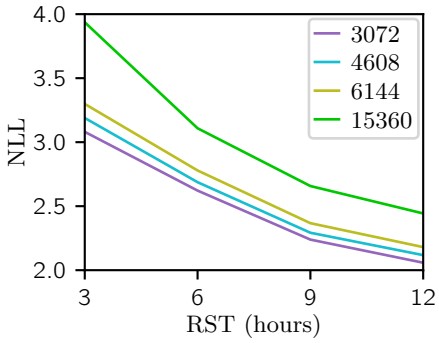

(a) RHO loss grid search: we tune the mega batch size hyper-parameter on BCWK as multiples of the mini-batch size (1536): {2x (3072), 3x (4608), 4x (6144), 10x (15360)}.

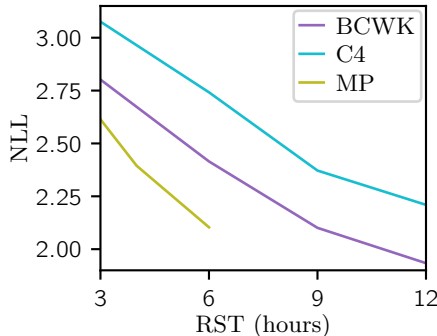

(b) Validation losses for training the RHO loss proxy/holdout model across three datasets.

We split all datasets into 20% proxy model pre-training, ~1% proxy model validation, and ~79% target model pre-training set (later, during target model pre-training, we further split the remaining 79% into train and validation set). For BCWK and C4, we train for 12 hours; for MP, we train for 6 hours only since it is much smaller and we want to avoid over-fitting. Other than varying the dataset splits and training budgets, we use the same hyper-parameters from Section 3.

Figure 10b shows the validation loss during proxy model pre-training; none of the models over-fit, and all of them reach reasonable losses. For reference, one may compare their values with the baseline in Figure 9, which shows the validation losses using data from the same dataset sources.

## C.5 Lion: Learning Rate (LR) And Weight Decay (WD)

The authors of Lion provide the following guidelines [13, page 14]:

> *"The default values for $\beta_1$ and $\beta_2$ in AdamW are set as $0.9$ and $0.999$, respectively, with an $\epsilon$ of $1e - 8$, while in Lion, the default values for $\beta_1$ and $\beta_2$ are discovered through the program search process and set as $0.9$ and $0.99$, respectively."*

We adopt these default $\beta_1, \beta_2$ hyper-parameters. Next, we look at LR and WD (also from [13]).

| Architecture | LR | WD |
|---|---|---|
| BERT-Base-like | {1e-4, 3e-4, 5e-4, 7e-4} | {0.03, 0.05, 0.07, 0.1} |
| T5-Base | {5e-4, 7.5e-4, 1e-3, 2.5e-3, 5e-3, 2e-2} | {0.0} |

Table 5: Grid Search Space for Lion.

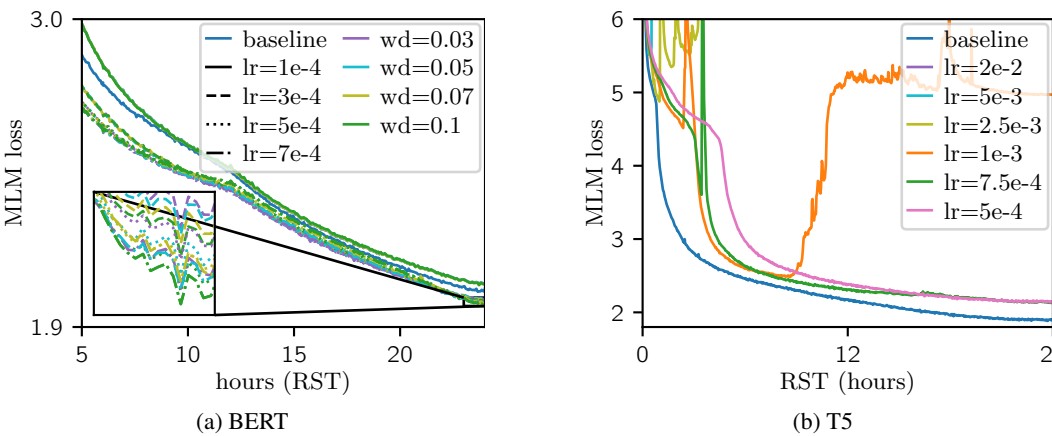

(a) BERT

(b) T5

Figure 11: Lion grid search for both the BERT and T5 model. Training loss over time during pre-training.

> *"Based on our experience, a suitable learning rate for Lion is typically 3-10x smaller than that for AdamW. Note that the initial value, peak value, and end value of the learning rate should be changed simultaneously with the same ratio compared to AdamW. We do not modify other training settings such as the learning rate schedule, gradient and update clipping. Since the effective weight decay is $\mathrm{lr} * \lambda$;* update $+= w * \lambda$; *update \*= lr, the value of $\lambda$ used for Lion is 3-10x larger than that for AdamW in order to maintain a similar strength."*

To recap (details in Section 3), for AdamW, we use base LRs of 1e-3 and 0.02, respectively. For BERT, we use a WD of 0.01, while we disable it for T5 respectively.

Hence, following the above guidelines, we define the grid search space as described in Table 5.

For BERT, we determine a LR of 7e-4 and a weight decay of 0.1 to be best, as illustrated in Figure 11a. For T5, we do not use any weight decay (following Raffel et al. [77], Shazeer [82]) and find an LR of 7.5e-4 to yield the best performance, as shown in Figure 11b. For all optimizers, we follow Nawrot [69] to integrate RMS-LR scaling to facilitate convergence.

## C.6  Sophia: Learning Rate (LR), Weight Decay (WD) and $\rho$

The official code repository [5] suggests the following:

---

[5] https://github.com/Liuhong99/Sophia/blob/main/README.md

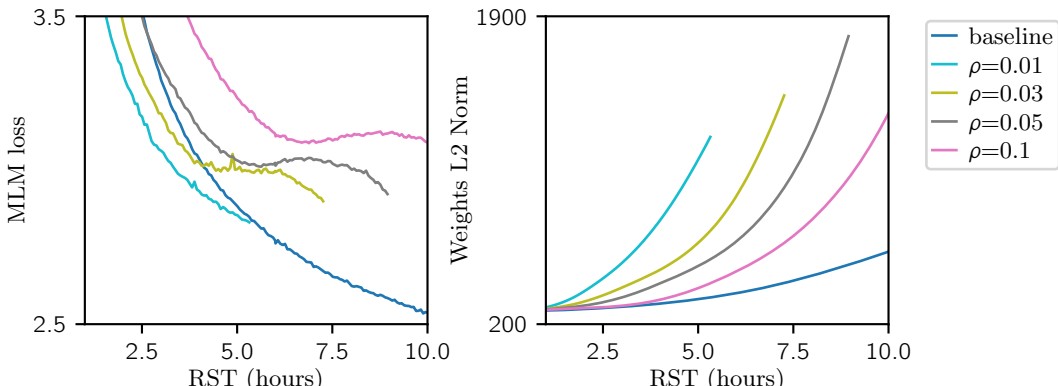

Figure 12: Inserting Sophia as Drop-in Replacement (FP16) for BERT resulted in NaN Losses.

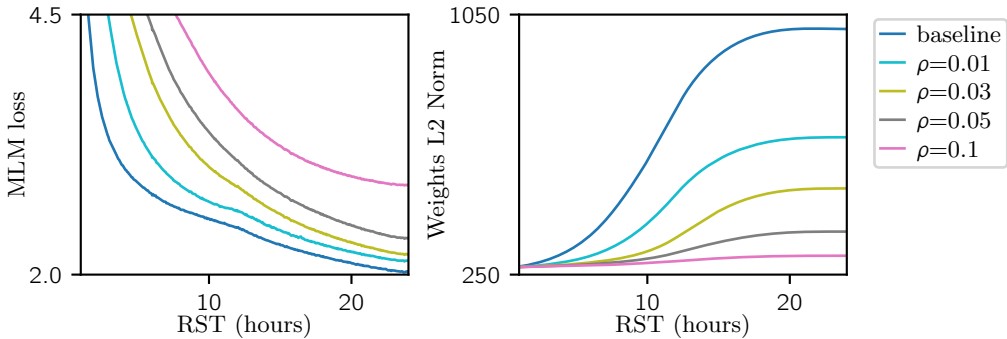

Figure 13: Lowering LR to 1e-4 of Sophia for BERT Slows Down Convergence (FP16).

> *"Choose lr to be about the same as the learning rate that you would use for AdamW. Some partial ongoing results indicate that lr can be made even larger, possibly leading to a faster convergence. Consider choosing $\rho$ in $[0.01, 0.1]$. $\rho$ seems transferable across different model sizes. We choose rho = 0.03 in 125M SophiaG. The (lr, rho) for 355M, Sophia-G is chosen to be $(5e - 4, 0.05)$ (more aggressive and therefore, even faster!) Slightly increasing weight decay to 0.2 seems also helpful."*

First, we followed the above guidelines, replaced AdamW with Sophia, and simply tuned $\rho \in \{0.01, 0.03, 0.05, 0.1\}$. All of these runs led to NaN losses, which we illustrate in Figure 12. Next, we lowered the learning rate to 1e-4, which mitigated the instabilities but resulted in much slower convergence, as shown in Figure 13.

We then decided to switch from FP16 to BF16, which improved training stability. Further, we manually tuned the LR, WD, and $\rho$ values, as shown in Figure 14. Since we noticed a strong negative correlation between $\rho$ and the training loss, we decided to stick to $\rho = 0.01$. The best performance was achieved with an LR of 4e-4 and a WD of 0.015. We follow Nawrot [69] to integrate RMS-LR scaling to facilitate convergence, as for all optimizers.

For T5, we vary the learning rate within the range of $\{2e-2, 5e-3, 2.5e-3, 1e-3, 7.5e-4, 5e-4\}$, and $\rho$ in $\{5e-2, 1e-2\}$. Figure 15 show the results; similar to BERT, we observe better performance with a smaller $\rho$. The best performance comes with $\rho = 1e - 2$ and LR of 1e-3.

We acknowledge that there are additional hyper-parameters like $\epsilon$ and $k$ that we did not tune because we followed the author's suggestions. Future work may investigate their effects on the training speed-ups too.

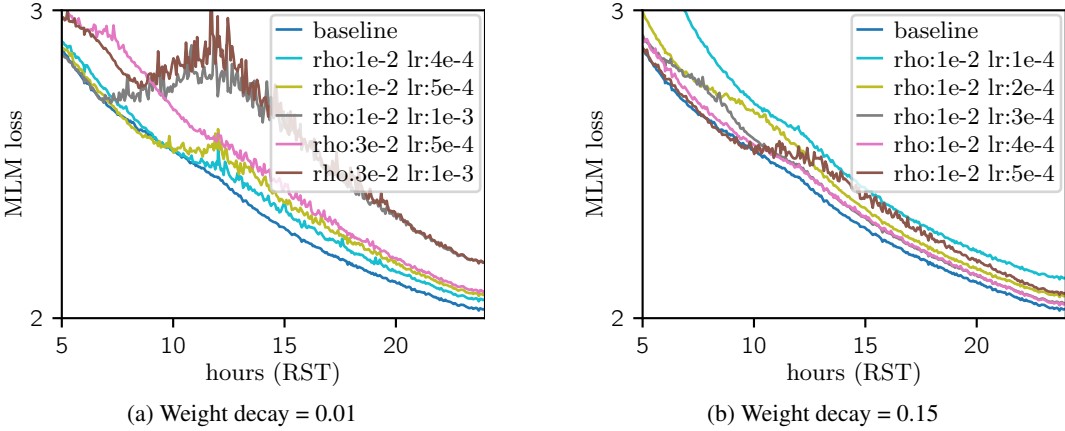

(a) Weight decay = 0.01                  (b) Weight decay = 0.15

Figure 14: Sophia grid search for BERT (BF16). Training loss over time during pre-training.

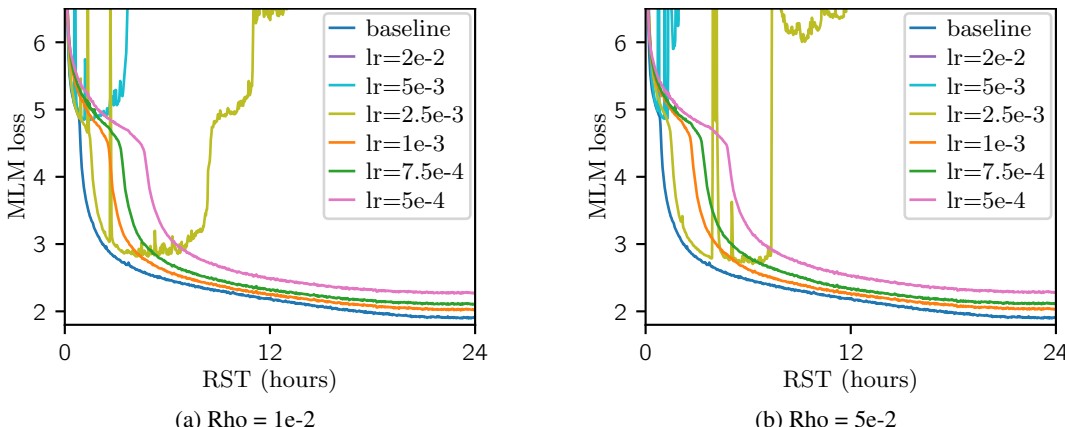

(a) Rho = 1e-2                       (b) Rho = 5e-2

Figure 15: Sophia grid search for the T5 model. Training loss over time during pre-training.

# D Additional Results

## D.1 Training BERT, Evaluating on GLUE

We notice that on specific GLUE datasets, the efficiency methods have little effect on performance (MNLI, SST-2, QNLI, QQP, MRPC). Across all datasets, CoLA appears to have the largest gains from efficient training. Averaged across all datasets, efficient training methods have little effect. We report the exact numbers in Table 1 (left) for layer stacking and layer dropping. In these experiments, layer dropping barely outperforms the baseline given a 6-hour RST budget. For a 12-hour budget, both stacking and layer dropping inch above the baseline, and stacking only produces more accurate results than the baseline and layer dropping for a 24-hour budget. In our study, we also compare the performance of the efficient methods on the T5-base model evaluated on the SNI benchmark and report the results in Table 2. From our observations, layer stacking is particularly noteworthy, demonstrating superior performance in a 6-hour training period, significantly outperforming the other methods. However, as the training budget is increased to 12 hours, the gap between baseline and layer stacking starts to diminish. In a 24-hour training scenario, baseline exhibits a marginally better performance than layer stacking, highlighting the efficacy of the baseline method given sufficient training time. Notably, both baseline and layer stacking consistently outperform layer dropping across all the time budgets.

## D.2 Layer dropping on small datasets

The layer dropping paper [107] trains on the Bookcorpus+Wikipedia dataset for 186 epochs, while the original BERT paper trains for only 40 epochs [22]. They do not report results for the baseline trained

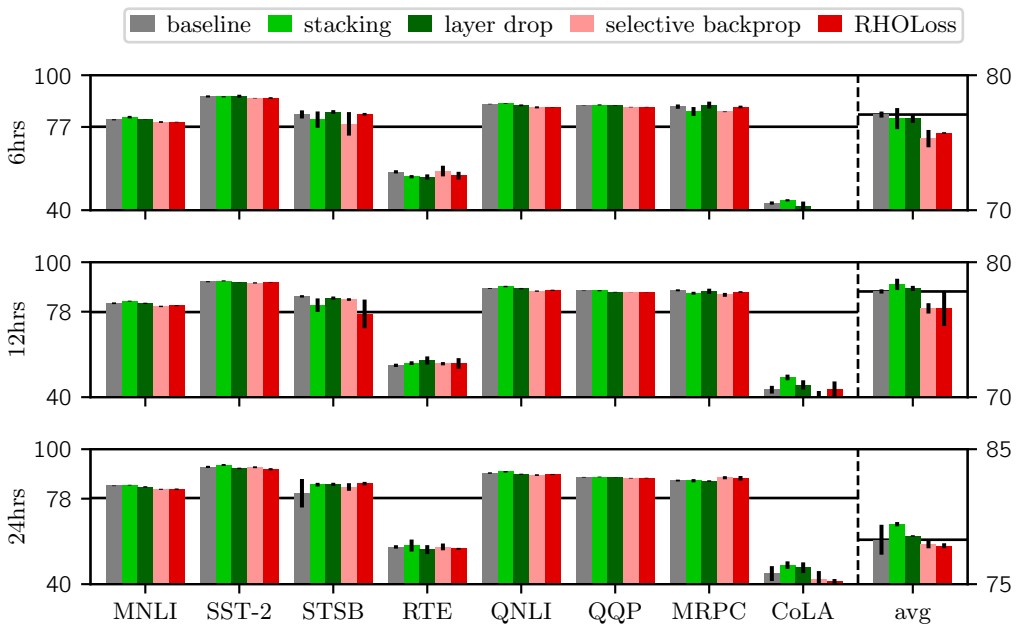

Figure 16: **BERT models evaluated on GLUE.** The black vertical error bars indicate the standard deviation over three seeds. The black horizontal line shows the average performance of the baseline. For clarity, the individual tasks are plotted against the left-hand axis, while the average accuracy is plotted against the right-hand axis.

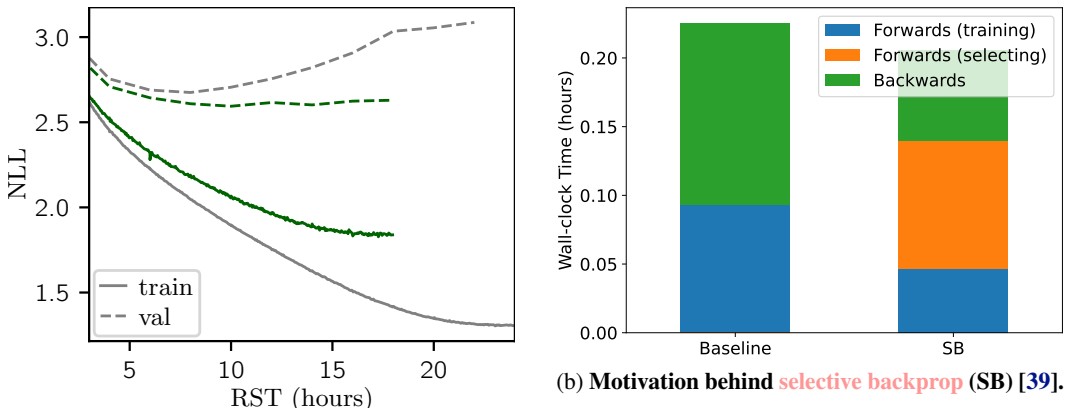

(a) **Ablation for layer dropping,** investigating its lack of performance. It prevents overfitting when performing multiple epochs over the dataset.

(b) **Motivation behind selective backprop (SB) [39].**

with a schedule based on fewer epochs. Given the possibility of overfitting due to the high number of epochs layer dropping's similarity to dropout [89], and dropout's known efficacy for language model pre-training with repeated data [104], we raise the question of whether layer dropping acts as a regularizer.

Note that given the abundance of pre-training data for language models, even the largest and most-expensively-trained models [34, 14, 92] are typically trained within a one-epoch regime, which has been shown to improve performance over training for multiple epochs on a smaller dataset [51, 77]. Hence, we exhaust the compute budget before passing through all available training data more than once. In contrast, Zhang & He [107] use a smaller dataset and thus complete 186 epochs during training. Thus, their training setup likely corresponds to an overfitting regime.

To investigate this, we repeat the experiment with a truncated training dataset, resulting in the experiment performing roughly 180 epochs. The result is shown in Appendix D.1. The plot confirms our suspicion: when the baseline training procedure overfits, layer dropping can help mitigate this, preventing the validation loss from increasing as training time increases.

