# A Hyper-Parameter Search

In this section, we showcase the hyper-parameter grid search we performed for layer stacking and selective backpropagation. For layer dropping, we include the search results in Figure 5a.

## A.1 Layer stacking: When To Stack

Figure 7 shows that layer stacking is relatively insensitive to different stacking RST hour times $\{(2.4, 7.2), (3, 7.2), (7.2, 9.6)\}$, with $\{(3, 7.2), (7.2, 9.6)\}$ performing about the same and $(2.4, 7.2)$ slightly worse. We choose $(3, 7.2)$, as it matches the same $\frac{\text{stacking step}}{\text{all training steps}}$ ratio as proposed by Gong et al. [27].

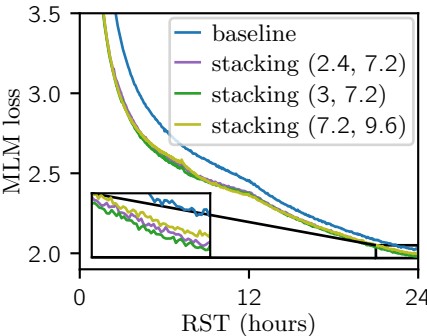

Figure 7: Layer stacking grid search: we tune the times at which the model is doubled. "stacking $(a, b)$" indicates that the model was doubled in size once at $a$ hours and then again at $b$ hours, measured using RST.

## A.2 Selective backpropagation: Selectivity Scale

We tune the selectivity scale $\beta$, where $\beta \in \{1, 2, 3\}$. Jiang et al. [35] use 33% and 50% selectivity in their experiments, which approximately corresponds to $\beta = \{1, 2\}$, respectively. We find that the larger the $\beta$ value, the worse the pre-training performance. Note that the higher the $\beta$ value, the more forward passes selective backpropagation needs to perform in order to collect enough samples for a backward pass, which decreases the total number of parameter update steps within the RST budget. For the experiments in Section 4.4, we chose $\beta = 1$, as it consistently achieves the best performance.

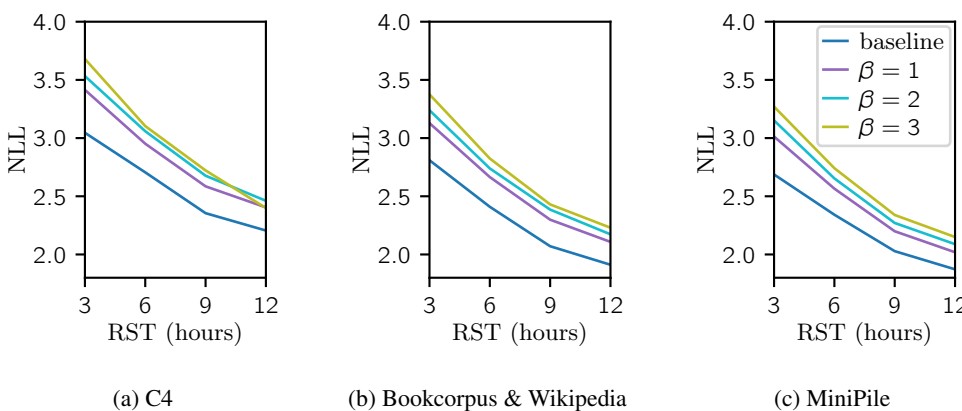

(a) C4       (b) Bookcorpus & Wikipedia       (c) MiniPile

Figure 8: Selective backpropagation grid search: we tune the $\beta$ hyperparameter. Each plot shows the validation loss over time during training for the given dataset.