# OpenReview forum: "No Train No Gain: Revisiting Efficient Training Algorithms For Transformer-based Language Models"
_NeurIPS.cc/2023/Conference — NeurIPS 2023 poster_

### Official Review · Reviewer_BpVn · 2023-06-29

**Soundness:** 3 good
**Presentation:** 2 fair
**Contribution:** 2 fair
**Rating:** 6
**Confidence:** 4

**Summary:**

This paper comprehensively evaluates three efficient training algorithms for transformer language models: layer stacking, layer dropping, and selective backpropagation. Such an evaluation is crucial due to the extensive computational resources needed for training transformer-based models.

The authors highlight the importance of specifying a training budget in performance comparisons. They introduce a measurement called "reference system time" (RST) to facilitate comparisons across different hardware configurations. The study examines these algorithms under various training budgets, model architectures, and datasets. The main finding is that these efficient training methods do not always significantly improve over the standard training baseline. Layer stacking tends to outperform the baseline and other methods in most instances. However, layer dropping sometimes falls short, while selective backpropagation often diminishes performance.

The paper also explores how these algorithms perform in a downstream task setting, showing that pre-training performance does not necessarily correlate with the ability to generalize in these tasks. It concludes by stressing the importance of caution when using these efficient training methods due to their potential overheads.  Further implications of this study can guide the choice of efficient training methods in transformer-based language models by considering the trade-off between improved performance and computational resources.

**Strengths:**

1. Originality: The paper's focus on evaluating efficient training algorithms for transformer language models under a defined training budget is a novel perspective. The introduction of "reference system time" to compare training times across different hardware configurations shows ingenuity.

2. Quality: The paper exhibits high quality in its methodology, experimental setup, and analysis. The experiments are thorough and well-conducted, involving different parameters like training budgets, model architectures, and datasets. The findings are meticulously analyzed, providing an honest assessment of the capabilities and limitations of the methods.

3. Clarity: The paper is well-written and logically organized. The authors clearly explain the motivations behind their work, the reason for choosing specific efficient training algorithms, and their experimental methodology. They discuss their findings in an understandable and straightforward manner.

4. Significance: The paper notably contributes to training language models. Its focus on efficient training methods and the introduction of RST offer practical solutions to the computational challenges researchers face. This research opens the door for more studies on efficient training of such models. The paper's findings and suggestions are invaluable for researchers and will aid in making informed decisions when implementing efficient training strategies.

**Weaknesses:**

1. Limited Scope: The evaluation in the paper is limited to three specific efficient training algorithms (layer stacking, layer dropping, and selective backpropagation).

2. Limited Novelty: This paper mainly focuses on revisiting methods in a specific context. There are not many novel ideas or groundbreaking discoveries introduced in the paper. Studies expected to be published in top-tier conferences like NeurIPS should ideally contain significant novel contributions.

3. Lack of Rigor in Experimental Design: While the authors propose using Reference System Time (RST), they do not sufficiently justify its superiority over other potential measures of computational effort.

4. Overemphasis on Computational Efficiency: The authors focused on computational efficiency, sidelining the quality or performance of the models. This imbalance in focus can give a skewed perspective.



**Questions:**

1. It would be beneficial to understand how the proposed Reference System Time (RST) handle variations in software optimization levels for different algorithms, which might influence the perceived efficiency of other methods.

2. For the RST measure, how much does the reference system time vary across different systems and configurations?

3. The paper asserts that the algorithms provided 'little effect' or 'marginal improvements'. Can you clarify what you consider to be substantial or significant improvements in efficiency and why the results obtained didn't meet that criteria?

4. In some comparison studies with baseline training, it's unclear whether the hyperparameters for the compared methods and the baselines were tuned independently. It would be beneficial if the authors clarified this aspect. Moreover, further details regarding the methodology behind establishing the "budget-adjusted baseline" would be appreciated.

**Limitations:**

The authors have clearly demonstrated some limitations in their paper, specifically highlighting that their study focuses on a select few efficient training algorithms and acknowledging that further exploration of additional algorithms is required. Furthermore, the authors admit that their review is primarily centered on language model pre-training and that the findings may not be necessarily applicable to fine-tuning or other data-intensive modalities.

However, the authors could have expanded on potential negative societal impacts. For instance, if these inefficient training algorithms are utilized for large-scale projects, it could lead to unnecessary usage of computing resources, subsequently affecting energy consumption and carbon emissions. Additionally, the misrepresentation of algorithm efficiency could potentially mislead the academic and industrial communities in their pursuit of efficient models.

To improve the paper, it would be beneficial for the authors to provide more context on possible wider implications, particularly focusing on the ecological impact and potential misdirection of research efforts. They might also consider suggesting some directions for future work in improving the efficiency of training algorithms, both in terms of reducing computational costs and minimizing environmental impact.

---

> ### Author Rebuttal · Authors · 2023-08-09
>
> > _1. Limited Scope_
>
> Thanks. Based on this feedback we have added three more recent efficient training algorithms. Please see the global response.
>
> > _2. Limited Novelty_
>
> Thank you for bringing this up. We argue that the observation that none of the popular recently-proposed efficient training algorithms improve upon a learning-rate-decayed baseline is actually ground-breaking for the efficient training field. Papers that make observations like this have been hugely impactful, for instance
>
> * [Are GANs Created Equal? A Large-Scale Study](https://proceedings.neurips.cc/paper/2018/file/e46de7e1bcaaced9a54f1e9d0d2f800d-Paper.pdf) discovered that most GAN architectures reach similar scores with enough hyperparameter optimization and random restarts
> * [In Search of Lost Domain Generalization](https://openreview.net/pdf?id=lQdXeXDoWtI) finds that the trivial baseline (ERM) outperformed most DG algorithms when evaluated under the same experimental conditions
> * [You CAN Teach an Old Dog New Tricks! On Training Knowledge Graph Embeddings](https://openreview.net/forum?id=BkxSmlBFvr) discovered that old techniques are competitive and often even outperform recent model architectures when the hyper-parameters are tuned correctly
> * [Rethinking the Value of Network Pruning](https://arxiv.org/abs/1810.05270), found that fine-tuning pruned models obtained by state-of-the-art structured pruning algorithms only give comparable or worse performance than training that model with randomly initialized weights
> * [Pitfalls of Graph Neural Network Evaluation](https://arxiv.org/abs/1811.05868), uncovered serious shortcomings in existing evaluation strategies for GNNs
>
> We actually started out trying to develop a new efficient training algorithm but then found that when we went to compare existing methods using a unified timing measure, reference system time (RST), none of the methods outperformed the simple baseline. Our shock at this discovery is what motivated us to publish this work. Thank you again for mentioning this, we will make this clearer in the final version.
>
> > _2. Lack of Rigor in Experimental Design_
>
> Thanks. Let us clarify the benefits of RST over the other most popular measures of compute: WCT and FLOPs. WCT can vary widely across different hardware, and can even fluctuate on the same hardware, for instance, due to the usage of non-deterministic operations, hidden background processes, or inconsequential configurations, such as the clock rate. We show an example of this in Figure 1, where we performed the same baseline training run across different hardware (a 3090 and an A100) and the same hardware but different software (different CUDA drivers on a 3090). FLOPs do not account for parallelism (e.g., RNNs vs Transformers) or hardware-related details that can affect runtime. For example, FLOP counting ignores memory access times (e.g., due to different layouts of the data in memory) and communication overheads, among other things.
>
> To address these things, RST ties all computation to a specific reference hardware. We average RST measurements over 1000 iterations (for each model architecture, batch, sequence length, system configuration, etc.), to ensure they are not influenced by fluctuations. In our code release, we will provide all RSTs measured by our hardware. This will enable researchers in the future to benchmark new algorithms using exactly our setup without owning the same hardware. Sorry this was not clearer, we will clarify this in the final version.
>
> > _3. Overemphasis on Computational Efficiency_
>
> We added training, validation, and downstream performance results in the rebuttal PDF. If there are any additional quality or performance metrics you would like to see we would be happy to add them.
>
> > Q1.
>
> Thanks for this question. We were curious about such fluctuations as well, to prevent them from influencing our results, we average RST measurements over 1000 iterations (for each algorithm, model architecture, batch, sequence length, system configuration, etc.). Thank you for bringing this up, we will clarify this in the final version.
>
> > Q2
>
> To see how much the timing on a reference system would change if we selected different systems as the reference, we can look at the WCT of individual systems (imagining each was the reference system). This is shown in the top set of WCT bars in Figure 1. In theory, any of the three systems could be the reference system, the bottom set of RST bars show what happens when the last system (the 3090, shown in dark red) is chosen as the reference system, and all other systems are mapped to the timing of this system. This ensures that the same baseline run is timed identically, regardless of the system it was run on.
>
> > Q3
>
> Thanks for this. By a `significant improvement’ we mean that, for a fixed RST budget, the average (training, validation, downstream) performance of an efficient training method must be greater than or equal to the average performance of the baseline plus a standard deviation (i.e., in Tables 1 and 2 we bolded the best method, and all methods whose average plus a standard deviation is greater than or equal to the best method). We will add these details to the final version.
>
> > Q4
>
> Thanks for this. In all experiments, the hyperparameters for each method were tuned independently. We added this to the supplement, but we can move it to the main paper if you think this would help. We will also add more details on our budget-adjusted baseline: the idea is that we train the full model (unlike layer stacking/dropping) using standard data loaders (unlike selective backprop/Rho-loss) using Adam (instead of Lion or Sophia). Further, we use a one-cycle learning rate schedule and adjust the learning rate schedule based on the elapsed time conditional on a time budget (measured in RST). The implementation for this is simple and will be included in our code release.

---

> > ### Comment · Reviewer_BpVn · 2023-08-19
> > **Thank you, concerns have been properly addressed.**
> >
> > The authors have addressed my concerns in a satisfactory manner. They have added three more recent efficient training algorithms to broaden the scope of the evaluation. They have also emphasized the ground-breaking nature of their observation that none of the popular recently proposed efficient training algorithms improves upon a learning-rate-decayed baseline. The authors have provided further clarification on the benefits of using Reference System Time (RST) as a measure of computational effort, addressing the concerns about fluctuations and variations in software optimization levels. They have also explained their definition of "significant improvement" and mentioned that the hyperparameters for each method were tuned independently. Additionally, the authors have provided more details on the methodology behind the "budget-adjusted baseline" and its implementation. Overall, my concerns have been properly addressed.

---

### Official Review · Reviewer_VTeA · 2023-07-03

**Soundness:** 3 good
**Presentation:** 3 good
**Contribution:** 3 good
**Rating:** 6
**Confidence:** 4

**Summary:**

Many algorithms have been proposed to make the training of ever larger models more efficient.
The authors present a critical empirical study of three selected training algorithms (layer stacking, layer dropping, and selective backpropagation) with fixed training budgets and find that these algorithms often do not make BERT and T5 pre-training significantly more efficient.

**Strengths:**

This paper re-visits evaluation standards, and offers a careful, rigorous, and clean re-evaluation of three efficient training algorithms.
It is an excellent example for the importance of also publishing "negative" results (no gains in metrics, no new algorithm).
Specifically, this finding can save development time in the implementation and maintenance of these algorithms and reduce the complexity of the pre-training algorithms.


**Weaknesses:**

I agree on the limitations pointed out by the authors in the section "Limitations and Future Work", including evaluation of small subset of efficient training algorithms only, language model pre-training only.
Overall, the paper might be felt to be too simple and straight forward, the more as it addresses a well-known issue in a noisy field.


**Questions:**

I agree with the authors that "As illustrated in Section 5, there is an abundance of efficient training algorithms, and rigorously evaluating all of them is prohibitively expensive." (Section 6). What would be an alternative approach to avoid such explosion of algorithms and experiments?

**Limitations:**

Yes, see the section "Limitations and Future Work".

---

> ### Author Rebuttal · Authors · 2023-08-09
>
> > I agree on the limitations pointed out by the authors in the section "Limitations and Future Work", including evaluation of small subset of efficient training algorithms only, language model pre-training only. Overall, the paper might be felt to be too simple and straight forward, the more as it addresses a well-known issue in a noisy field.
>
> Thank you for this. Our motivation for evaluating a subset of efficient training algorithms on language models only was to allow us to spend more careful detail on an in-depth analysis of these methods. We believe the field of efficient training methods for language models will grow rapidly in the coming years, as the most successful models can currently only be trained by a handful of entities.
>
> As far as we know, we haven’t seen any paper comparing the main approaches in this field. Our goal was to fill this gap to help researchers and practitioners understand which approaches deserve more attention and to spur development in these directions. We were surprised to discover that, in large part, current efficient training algorithms do not surpass learning-rate-decayed baseline training. We believe these observations could act as a turning-point for the efficient training field and prompt investigation into new directions entirely. Thank you for bringing this up; we will add more discussion on this in the final version.
>
> > I agree with the authors that "As illustrated in Section 5, there is an abundance of efficient training algorithms, and rigorously evaluating all of them is prohibitively expensive." (Section 6). What would be an alternative approach to avoid such explosion of algorithms and experiments?
>
> This is an interesting open question. The approach we took is to evaluate representative algorithms on popular architectures, using well-studied tasks. We believe our approach is particularly useful as the field continues to grow because, by using our published RST timings, it will allow future algorithms, architectures, and tasks to be evaluated against our results, regardless of the hardware setup.

---

> > ### Comment · Reviewer_VTeA · 2023-08-17
> >
> > Thank you for the response. I largely agree with iTEM's "Thanks for your response" comment.

---

### Official Review · Reviewer_iTEM · 2023-07-04

**Soundness:** 3 good
**Presentation:** 3 good
**Contribution:** 3 good
**Rating:** 7
**Confidence:** 3

**Summary:**

The paper reevaluates several training algorithms aimed at enhancing the efficiency of Transformer-based models, such as layer stacking, layer dropping, and selective backpropagation. The authors effectively manage the training resources by employing a metric called reference system time. However, the key result of the study indicates that these three efficient training algorithms yield only modest improvements compared to standard training methods.

**Strengths:**

- The authors conduct a thorough evaluation of efficient training algorithms, providing valuable insights into their effectiveness and highlighting the marginal gains such methods achieve. This revisit contributes to a deeper understanding of existing methods and offers a practical perspective on their applicability.
- The authors point out the shortcomings of using wall-clock time for reference, and instead propose reference system time (RST) for a better estimate, which is valuable rule to highlight.

**Weaknesses:**

- While revisiting these methods undoubtedly holds value for the research community, it is important to note that the obtained results may be somewhat trivial and lack significant insights. The paper might be better suited for more specialized venues.
- While the primary focus of the study lies in evaluating the original approaches, providing additional discussions on methods and evaluations that build upon these evaluated techniques would offer a more comprehensive understanding of how these methods have been adopted and evolved since their initial proposal.

**Questions:**

Do the results of this study depend on the size of the models? Can we anticipate that the effectiveness of these efficient training techniques would be more diminished or enhanced when training larger models with a greater compute budget?

**Limitations:**

In Section 6, the authors acknowledge the limitations of their work and highlight that it is not feasible to evaluate all types of training efficient algorithms due to the potentially exorbitant costs involved.

---

> ### Author Rebuttal · Authors · 2023-08-09
>
> > While revisiting these methods undoubtedly holds value for the research community, it is important to note that the obtained results may be somewhat trivial and lack significant insights. The paper might be better suited for more specialized venues.
>
> Thanks for this. We have added three more recent efficient training algorithms. Please see the global rebuttal response.
>
> > While the primary focus of the study lies in evaluating the original approaches, providing additional discussions on methods and evaluations that build upon these evaluated techniques would offer a more comprehensive understanding of how these methods have been adopted and evolved since their initial proposal.
>
> Thanks for this, we will add a discussion on methods and evaluations that build on the efficient training algorithms we compare. On this note, the new methods we have added are motivated as building on the prior work we have compared here. Specifically, Sophia [(Liu et al., 2023)](https://arxiv.org/pdf/2305.14342.pdf) builds upon Lion [(Chen et al., 2023)](https://arxiv.org/pdf/2302.06675.pdf), as [Liu et al., 2023](https://arxiv.org/pdf/2305.14342.pdf) claim that Lion “only achieves limited speed-up on LLMs“. RHO-Loss [(Mindermann et al., 2022)](https://proceedings.mlr.press/v162/mindermann22a/mindermann22a.pdf) builds on [selective backprop](https://arxiv.org/pdf/1910.00762.pdf), as [Mindermann et al., 2022](https://proceedings.mlr.press/v162/mindermann22a/mindermann22a.pdf) argue that solely prioritizing high training loss results in two types of examples that are unwanted: (i) mislabeled and ambiguous data, as commonly found in noisy, web-crawled data; and (ii) outliers, which are less likely to appear at test time. Thank you for bringing this up, we will add these things in the final version.
>
> > Do the results of this study depend on the size of the models? Can we anticipate that the effectiveness of these efficient training techniques would be more diminished or enhanced when training larger models with a greater compute budget?
>
> Great question! Our BERT model has 120M parameters and T5 has 247M, and across these model sizes the results appear to be consistent. We suspect the effect is similar for other LM sizes. That said, if there is another model or model size you would like to see, let us know, and we would be happy to include it.

---

> > ### Comment · Reviewer_iTEM · 2023-08-17
> > **Thanks for your response**
> >
> > I read the author's responses carefully and believe that there is tremendous value in publishing these negative results, especially when they carry significant implications for future model training. I have raised my score to 5. However, I believe the review should focus on the original submission rather than newly added techniques. Thus I would not be able to further increase the score.

---

> > > ### Comment · Senior_Area_Chairs · 2023-08-20
> > > **It is OK to increase your score further if you want.**
> > >
> > > Just quickly commenting that you do not have to limit your score to only focus on the original submission. If the rebuttal changed your opinion, you can make a bigger update, but of course do not have to.

---

### Official Review · Reviewer_DsDW · 2023-07-04

**Soundness:** 2 fair
**Presentation:** 2 fair
**Contribution:** 3 good
**Rating:** 6
**Confidence:** 5

**Summary:**

This paper studies three efficient pertaining techniques for Transformer models (layer stacking, layer dropping, and selective backpropagation). At variance with previous works on the subject, the authors adequately control for the learning rate schedule by evaluating performance at fixed compute budgets (as defined through a scaled wall-clock time). In doing so, they find that most of these methods provide unclear gain over a vanilla baseline -- and this finding holds across T5/BERT-like models.

**Strengths:**

* **S1.** At variance with previous works, the authors adequately control for the effect of the learning rate schedule. This is a common mistake in countless papers, which should be more discussed as it has previously impacted prominent work around scaling laws (see Hoffmann et al., 2022 as a response to Kaplan et al., 2020).

* **S2.** This paper provides negative results on the timely topic of efficient pretraining of Transformers, allowing the community to take a step back and retrospectively analyse the validity of previous findings. Under that light, negative results can be as valuable as positive ones.

* **S3.** (minor) The color scheme (which is consistent across text & plots) is a good idea to visually help readers.

**Weaknesses:**

* **W1. Limited significance due to limited adoption of the practices evaluated.** The impact and significance of this work is limited, as it evaluates three methods which have not been widely adopted by the community. For a negative result paper to be truly valuable, it is better if it addresses a common practice rather than a result that has not gotten traction anyway.


* **W2. It is unclear how the proposed Reference System Time approach improves upon other methods.** The authors bring-up RST as one of the significant contribution of the paper, and base their analysis on its usage. However, it's unclear how it differs in practice from simply using wall-clock time -- especially since the authors don't actually perform cross-hardware comparisons, which would be one of the benefit of formalising RST.
   * **W2.1** The authors claim "Unfortunately, WCT can fluctuate even on the same hardware, for instance, due to the usage of non-deterministic operations, hidden background processes, or inconsequential configurations, such as the clock rate". It is not clear how RST improves upon this -- the first recording used to scale RST may be noisy as well, and so could the recordings being scaled.
   * **W2.2** It is unclear whether Figure 1 is based on real measurements or is simply there for illustrative purpose.


* **W3. The results are disparate, incomplete, and lack clarity.** The results are middling, lacking a clear finding (overall it seems layer dropping is worth it across all budgets for BERT, layer stacking may bring gains for longer training budgets on BERT but only on shorter budgets for T5, and that selective back propagation is never worth it). For a paper with negative results, clarity is key to improve upon past work.
    * **W3.1.** Selective back propagation is compared in a completely different setup, on a single budget, and downstream performance is never measured for SBP+T5 -- only validation loss. Furthermore, while SBP+BERT are trained on 3 different datasets (a rather interesting study given the nature of SBP), only validation losses are provided for comparisons (no downstream task performance) and no conclusion is proposed on the influence of data source on SBP. The lack of a unified framework for comparisons between layer stacking/dropping and SBP make the paper feel more like a pot pourri of ideas rather than a principled & systematic comparison.
    * **W3.2.** Reporting training loss instead of validation loss in Figure 2 is questionable: as the authors later discuss in their ablations in Figure 5.b), layer dropping acts similarly to drop out -- so it makes sense it is behind on training loss in Figure 2. Validation loss here would be far more valuable.
    * **W3.3.** The figures are not clear: the x-axis in Figure 3 is not specified, the legends lack an analysis/mention of the key results for each figure, it's impossible to see differences in Figure 4, etc. Specifically to Figure 4/Table 1, the authors mention the concept of Pareto front multiple times in the main text; maybe plotting it would be far clearer than the figures/table proposed for downstream evaluation.


* **W4. Insufficient clarity and lack of added educational value.** Negative results papers can prove themselves especially valuable if they help correct a bad practice in the community; here, a general lack of clarity and of adequate framing make this paper fall short of being an educational read.
   * **W4.1.** The introduction of RST is confusing and frankly not necessary, as discussed in W2.
   * **W4.2.** The issue around learning rate schedules could benefit from additional discussion & illustration. In particular, it would be interesting to mention that this is the crux of the difference between the scaling laws of Kaplan et al., 2020 and Hoffman et al., 2022. Not accounting for the influence of the LR schedule caused the Kaplan work to inappropriately recommend scaling model size primarily, instead of scaling model and data jointly.
   * **W4.3.** The paper contains numerous imprecisions and overall lacks clarity (see W3.3. as well). The authors sometime abuse the citations by citing far too many work at once instead of the most relevant ones (l14, l32, l78, etc.). The default citation style of NeurIPS doesn't help here, making it difficult to identify the works at a glance... l144 it's unclear for instance what the sentence of curriculum and deduplication brings to the work -- since this never further discussed. Note also that Figure 6 is never referenced in the main text, and that "Besides the dizziness due to conflicting ideas" on l30 is not adequate language for a paper.

**Questions:**

Although exploring an interesting and valuable premise, this paper is insufficiently impactful and held back by its somewhat unclear and unprincipled methodology in the results section. I would recommend for the authors to: (1) unify their methodology across layer stacking/dropping/selective backpropagation and all models/datasets, to deliver a clearer results section; (2) improve the presentation of their results, with clearer figures (potentially plotting Pareto fronts as proposed in the main text); (3) get rid of the RST idea, which adds confusion for no reason. In its current form, I would rate this paper as a **Reject (3)**.

**EDIT: following rebuttal I have updated my score to a Weak Accept (6)**.

**Q1.** Based on the comments in W2 above on RST, could the authors clarify the value they see in RST? Especially regarding smoothing out fluctuations.

**Q2.** Could the authors provide additional results for selective backpropagation at different compute budgets, and expand on the experiments combining SBP with different datasets?

**Limitations:**

The authors have included a limitations section.

---

> ### Author Rebuttal · Authors · 2023-08-09
>
> > _W1. Limited significance_
>
> Thank you for pointing this out. Based on this feedback, we have added three more recent efficient training algorithms, see the global rebuttal response.
>
> > _W2. unclear how the proposed Reference System Time approach improves_
>
> We believe there is a small confusion here. The difference between WCT and RST is that WCT does not account for changes in hardware and software configurations. RST does this by mapping all timings on an arbitrary system back to the reference system. We show an example of this in Figure 1, where we performed the same baseline training run across different hardware (a 3090 and an A100) and different software (different CUDA drivers on a 3090). And in fact, we ran all of our experiments across different machines (3090s and A100s), which enabled us to run such a large number of experiments (including hyer-parameter tuning, etc.). Sorry this was not clearer, we will make this clear in the final version.
>
> > _not clear how RST improves upon this_
>
> Thanks for this. To ensure our RST measurement is not prone to fluctuation, we averaged these timings over 1000 iterations (for each model architecture, batch, sequence length, system configuration, etc.). We put this in a footnote, but we will move this to the main text to highlight this.
>
> > _W2.2 It is unclear whether Figure 1 is based on real measurements_
>
> Figure 1 is based on real measurements (the time of a baseline training run). We will make this explicit in the final version, thanks!
>
> > _W3. The results are disparate, incomplete, and lack clarity._
>
> Thank you for pointing that out. To improve clarity and incomplete results, we ran additional experiments and summarized our main findings; both can be found in the global response.
>
> > _W3.1. Selective back propagation is compared in a completely different setup_
>
> Thanks. Based on this suggestion, we added validation and downstream performances for both SBP and Rho-Loss for all three considered budgets and a new downstream benchmark (SuperGLUE) to the new rebuttal PDF. We’ve also investigated the influence of different training data sources on both SBP and Rho-Loss in (Table 5 and Figure 4, rebuttal PDF). We opted to spend compute on this instead of on additional T5 experiments, as the effects here are large for batch selection methods. Across three training datasets, none of the batch selection methods outperform the validation loss or downstream performance of the baseline.
>
> > _The lack of a unified framework for comparisons between layer stacking/dropping and SBP make the paper feel more like a pot pourri of ideas._
>
> We have now included downstream comparisons for all methods in the rebuttal PDF (in Tables 1 and 2, Figures 1 and 2). We compared SBP and Rho-Loss with the baseline using validation loss instead of training loss as these methods intentionally select high-loss training batches to improve generalization. Across three training datasets, neither SBP or Rho-Loss outperformed the validation loss or downstream performance of the baseline (Tables 3, 5; Fig. 4 in rebuttal PDF). If we don’t count the time required to select batches, RHO-Loss can slightly improve the validation loss (Table 4, rebuttal PDF).
>
> > _W3.2. Reporting training loss instead of validation loss in Figure 2 is questionable._
>
> Thank you for bringing this up. We report training loss because we sample without replacement from our training dataset (C4), which is so large that we never see the same data-point twice throughout training. This means that the training loss is always computed on inputs that have not been seen before and so is an unbiased estimate of the generalization error, just like the validation loss.
>
> Also, previously this figure reported the training loss computed while the layers were dropped. In the rebuttal PDF, we have included a new version of the figure where we compute the loss with all layers enabled. This does not qualitatively change the interpretation of the figure.
>
>  > _W3.3. The figures are not clear._
>
> Thank you! This should be the training budget. We have added this to the rebuttal PDF (Figure 4). This figure shows that the efficient batch selection algorithms do not outperform the baseline regardless of the training dataset.
>
> > Specifically to Figure 4/Table 1, the authors mention the concept of Pareto front multiple times in the main text; maybe plotting it would be far clearer than the figures/table proposed for downstream evaluation.
>
> We believe there is a slight confusion here, these are in fact points on the Pareto curve of downstream performance vs. time (RST for 6, 12, 24 hours). The reason we opted for bar plots and tables is because all of the methods are essentially overlapping each other.
>
> > W4. Insufficient clarity and lack of added educational value.
>
> Based on your prior comments, we have revised the manuscript to improve clarity (unfortunately, we cannot upload revised manuscripts in the rebuttal phase this year) and scope (additional experiments in rebuttal PDF). If you have any other recommendations, we would be happy to implement them.
>
> > _W4.1. The introduction of RST is confusing_
>
> See our response to W2 above.
>
>  > _W4.2. The issue around learning rate schedules could benefit from additional discussion._
>
> This is a very nice suggestion, we will investigate this further and add a discussion on this to the final version.
>
> > _W4.3. The paper contains numerous imprecisions_
>
> We will revise our manuscript and (1) compress citations to the most relevant ones, (2) explain the background of sentence curriculum and deduplication in the pretraining dataset, (3) cite Figure 6 in the main text, and (4) we will rewrite the paragraph starting with "Besides the dizziness due to conflicting ideas". We hope this addresses your concerns, and we welcome any more suggestions on the writing.
>
> > Q1.
>
> Yes, please see the response to W2 above.
>
> > Q2.
>
> Yes, we have included these additional results in the rebuttal PDF.

---

> > ### Comment · Reviewer_DsDW · 2023-08-16
> > **Answer to rebuttal**
> >
> > First, I would like to note that I appreciate that the authors have written-up an extensive rebuttal to all reviewers, with significant additional results. The scope of the changes makes it difficult to evaluate the paper in its entirety again, but I won't blame the authors on this and take this as a quirk of NeurIPS allowing a 1-page results .pdf rebuttal for the first time in a while.
> >
> > The authors have adequately addressed some of my concerns: W1. (the added methods are interesting and paint a broader picture), W3., and likely W4. as well.
> >
> > However, I would maintain W2. (related to doubts around the introduced concept of RST). To better explain my concern, I simply think that a lot of lines are "wasted" on what is essentially a common idea of "comparing things that are comparable" / "rescaling". The authors highlights that the rescaling allows to compare across hardware and drivers version, but I seriously question why you would even consider doing this in the first place. This sounds like an overcomplexification of the evaluation setup employed.
> >
> > With that in mind, I will update my score to a **Weak Accept (6)** based on the extensive update proposed by the authors.

---

> > > ### Author Response · Authors · 2023-08-16
> > > **Thank you so much! Further comments on W2.**
> > >
> > > Dear Reviewer DsDW,
> > >
> > > Thank you so much for your time and for reconsidering your score.
> > >
> > > On W2, we agree with you that measuring RST is a simple idea, and we will revise our transcript to make sure that not too many lines are spent on over-selling it.
> > >
> > > The motivation for why one would even consider rescaling in the first place is that it is necessary to perform the large number of pre-training runs, in this and similar papers, in parallel to obtain results in a reasonable amount of time. This leads to many scenarios where WCT can differ from run to run, for example:
> > >
> > >
> > >
> > > 1. **Single nodes containing multiple GPUs**. If multiple jobs are running on individual GPUs on a single node, despite the software and hardware being fixed, the utilization of the shared resources (eg., CPU, RAM, Disk) can influence the job completion time. For example, [[1](https://onlinelibrary.wiley.com/doi/pdf/10.1002/cpe.6730?casa_token=NQSh5k_gkMoAAAAA:ziA4HNyHhbfsEFwdKuB220z4s73xVPH6He9JC7eT4wtFsAs0Gg92mfS39pBeo2ovRaOfOIv-CiCNIwA)] discusses the CPU and RAM utilization of data loading; the more jobs one runs, the more bottlenecked these operations become and the longer the completion time.
> > > 2. **Multiple on-premise nodes** with slightly different configurations (different hardware, software, etc.) because they were bought at different times, with different budgets, or are maintained by different people. For example, we looked at the [best paper award papers from NeurIPS last year](https://nips.cc/virtual/2022/awards_detail) and found three which used inconsistent hardware throughout experiments (which is OK in their case since they were not concerned about WCT): [[2] ](https://arxiv.org/pdf/2206.14486.pdf)(NVIDIA TITAN Xp and V100 GPUs), [[3]](https://proceedings.neurips.cc/paper_files/paper/2022/file/27c546ab1e4f1d7d638e6a8dfbad9a07-Paper-Conference.pdf) (NVIDIA RTX A5000 and Quadro RTX 8000 GPUs), [[4](https://proceedings.neurips.cc/paper_files/paper/2022/file/105112d52254f86d5854f3da734a52b4-Supplemental-Conference.pdf)] (GeForce RTX 1080, 1080 Ti and 2080 Ti GPUs).
> > > 3. **Shared compute cluster** (potentially provided by a cloud provider). Such systems are fairly complex, and fluctuations in job completion times (even for identical jobs) are fairly common. For example, Figure 11 from [[5]](https://www.usenix.org/system/files/nsdi22-paper-weng.pdf) shows this happening inside a production MLaaS cluster with over 6,000 GPUs in Alibaba. More generally, mitigating job interferences due to resource sharing in GPU clusters is an open research question in itself, see eg. Section 4.2.1 in [[6]](https://arxiv.org/pdf/2205.11913.pdf).
> > >
> > > We hope this clarifies your concerns on W2. Thank you very much for your detailed feedback.

---

### Official Review · Reviewer_b1c1 · 2023-07-10

**Soundness:** 3 good
**Presentation:** 3 good
**Contribution:** 2 fair
**Rating:** 5
**Confidence:** 4

**Summary:**

The paper presents an analysis of 3 algorithms for training transformer models with a focus on efficiency. The authors present a way to measure wall clock time irrespective of the underlying hardware and make a principled comparison of the 3 algorithms by predefining a compute budget and adapting each algorithm and training recipe for the available time. The results of the paper show that none of the methods is universally an improvement over standard training.

**Strengths:**

- Meta-analyses are very common in other fields but are sorely lacking in ML/Deep learning so the goal of this work is very welcome and in my humble opinion useful to the community.
- Setting a compute budget and trying to optimize each algorithm accordingly is not as common as it should be in similar works.

**Weaknesses:**

- The main weakness of the work is possibly the choice of algorithms to be analyzed. There are only 3 while there could be a plethora of others as mentioned in the paper's related work. Moreover, the 3 methods evaluated are not particularly widespread (possibly because they don't work as well as the paper shows).
- Another possible weakness is the fact that there is separate hyper parameter tuning per method irrespective of the compute budget. This means that the results could be widely different were one to apply the above methods to a new problem where the optimal or near-optimal hyper parameters are not known a priori.
- RST although fair, it is not a transferable metric which means that a different paper will not be able to compare with the RST results of this paper and will have to re-run the experiments to compute the relative speed of the methods on their hardware. This severely diminishes its usefulness.

**Questions:**

My main reservation for the paper is whether the experiments are enough to prove useful to the community.

**Limitations:**

The authors properly present and analyze the limitations of their work.

---

> ### Author Rebuttal · Authors · 2023-08-09
>
>
> > _The main weakness of the work is possibly the choice of algorithms to be analyzed. There are only 3 while there could be a plethora of others as mentioned in the paper's related work. Moreover, the 3 methods evaluated are not particularly widespread (possibly because they don't work as well as the paper shows)._
>
> Thank you for pointing this out. Based on this feedback, we have added three more recent and popular efficient training algorithms from this plethora of other methods mentioned in the related work section. To better organize all methods we have separated them into three different types: (1) **dynamic architectures** (i.e., the existing [layer stacking](http://proceedings.mlr.press/v97/gong19a/gong19a.pdf)/[dropping](https://proceedings.neurips.cc/paper_files/paper/2020/file/a1140a3d0df1c81e24ae954d935e8926-Paper.pdf) methods), (2) **batch selection**, where we add RHO-Loss [(Mindermann et al., 2022)](https://proceedings.mlr.press/v162/mindermann22a/mindermann22a.pdf) (and the existing [selective backprop](https://arxiv.org/pdf/1910.00762.pdf) method) , and (3) **efficient optimizers**, where we add recent optimizers Lion [(Chen et al., 2023)](https://arxiv.org/pdf/2302.06675.pdf) and Sophia [(Liu et al., 2023)](https://arxiv.org/pdf/2305.14342.pdf). The three new methods are very popular, e.g., their GitHub repositories have been highly starred: [1600 for Lion](https://github.com/lucidrains/lion-pytorch), [700 for Sophia](https://github.com/Liuhong99/Sophia), and [161 for Rho-Loss](https://github.com/OATML/RHO-Loss), and have been released recently (Sophia was released on May 25th, 2023; Lion on February 15th, 2023; RHO-Loss on June 16th, 2022).
>
> Please let us know if there are other methods you would like to see tested.
>
> > _Another possible weakness is the fact that there is separate hyper parameter tuning per method irrespective of the compute budget. This means that the results could be widely different were one to apply the above methods to a new problem where the optimal or near-optimal hyper parameters are not known a priori._
>
> Thanks for this. We agree that placing a budget on hyperparameter tuning is important. However, if we also placed a budget here, one could argue that the reason the baseline outperforms efficient training algorithms is because the baseline has fewer hyperparameters to tune, and so better hyperparameters can be found compared to other methods under this fixed budget. To avoid this critique, we fully tuned all efficient training algorithms, making them as strong as possible. We expect that in new problems with a fixed budget on training and hyperparameter tuning, the performance of these efficient training algorithms will degrade even further w.r.t. the baseline. Thank you again for bringing this up, we will clarify this point in the final version.
> > _RST although fair, it is not a transferable metric which means that a different paper will not be able to compare with the RST results of this paper and will have to re-run the experiments to compute the relative speed of the methods on their hardware._
>
> We believe there is a slight confusion here. We saved all timings on the reference system for the model architectures, batches, sequence lengths, and other configurations used in our experiments. We will release these timings, allowing other researchers to run experiments with the same RST budget as computed against this reference system. This will prevent them from having to rerun any experiments in our paper. Sorry that this was not clearer, we will clarify this in the final version.

---

### Author Rebuttal · Authors · 2023-08-09

We would like to thank all reviewers for their insightful and encouraging reviews:
* **b1c1** - _“the goal of this work is very welcome and in my humble opinion useful to the community_”;
* **DsDW** - “_allowing the community to take a step back and retrospectively analyse the validity of previous findings_”’;
* **iTEM** - “_[it] contributes to a deeper understanding of existing methods and offers a practical perspective on their applicability_”;
* **VTeA** - “_It is an excellent example for the importance of also publishing "negative" results (no gains in metrics, no new algorithm)._”;
* **BpVn** - “_The paper's findings and suggestions are invaluable for researchers and will aid in making informed decisions when implementing efficient training strategies._”.

In response to requests for comparing more methods (**b1c1, DsDW**), we added three more recent efficient training algorithms and one more benchmark ([SuperGLUE](https://arxiv.org/abs/1905.00537)). The results can be found in the attached rebuttal PDF.

To better categorize all approaches, we have separated the methods into three different types: (1) **dynamic architectures** (i.e., the existing [layer stacking](http://proceedings.mlr.press/v97/gong19a/gong19a.pdf)/[dropping](https://proceedings.neurips.cc/paper_files/paper/2020/file/a1140a3d0df1c81e24ae954d935e8926-Paper.pdf) methods), (2) **batch selection**, where we add RHO-Loss [(Mindermann et al., 2022)](https://proceedings.mlr.press/v162/mindermann22a/mindermann22a.pdf) (and the existing [selective backprop](https://arxiv.org/pdf/1910.00762.pdf) method), and (3) **efficient optimizers**, where we add recent optimizers Lion [(Chen et al., 2023)](https://arxiv.org/pdf/2302.06675.pdf) and Sophia [(Liu et al., 2023)](https://arxiv.org/pdf/2305.14342.pdf). The three new methods are very popular, e.g., their GitHub repositories have been highly starred: [1600 for Lion](https://github.com/lucidrains/lion-pytorch), [700 for Sophia](https://github.com/Liuhong99/Sophia), and [161 for Rho-Loss](https://github.com/OATML/RHO-Loss), and have been released recently (Sophia was released on May 25th, 2023; Lion on February 15th, 2023; RHO-Loss on June 16th, 2022). We have included a one-page PDF with new experimental results for all methods.

For the new optimizers, we noticed numerical instabilities when adding them as drop-in replacements into our training pipeline. Changing the mixed precision helped, which is why we report their results in BF16 instead of FP16. We also re-ran the baseline in that new precision to ensure direct comparability.

Our main findings are:



* **Training loss** (comparing: Layer stacking, Layer dropping, Lion, Sophia (for **batch selection** approaches: selective backprop and RHO-Loss, we instead compare the validation loss, as they intentionally select training batches with high loss)): The only approach to consistently outperform the training loss of the fully-decayed learning rate baseline across budgets and models is Layer stacking (see Figure 1 in the rebuttal PDF). This improvement reduces as the budget increases to 24 hours.
* **Validation loss** (selective backprop, RHO-Loss): Across three training datasets, none of the **batch selection** methods outperform the validation loss of the baseline (Table 3 and Figure 4 in rebuttal PDF). If we don’t count the time required to select batches, RHO-Loss can slightly improve the validation loss (Table 4, rebuttal PDF).
* **Downstream tasks**: For a 24-hour budget, none of the efficient training algorithms we evaluate improves the downstream performance of the baseline (Tables 1 and 2, Figures 2 and 3, rebuttal PDF).
* Methods with lower per-iteration costs than the baseline (i.e., **dynamic architecture** methods: Layer stacking, Layer dropping) can slightly improve downstream performance for lower budgets (6 hours, 12 hours), but the improvement disappears with longer training.
* Methods with higher per-iteration costs (i.e., **batch selection** methods: selective backprop and RHO-Loss, and some **efficient optimizer** methods: Sophia) are significantly worse than the baseline in some downstream tasks (GLUE, SNI), for all budgets.
* If we ignore the additional per-iteration computations of the three above methods, the downstream performance is still matched by the baseline.

Thank you for your time spent on reviewing our work.

---

### Decision · Program_Chairs · 2023-09-21

**Decision:**

Accept (poster)

**Comment:**

The submission compares the performance of recently proposed improvements to transformers. It aims to provide a fairer comparison than previous work by introducing the concept of 'reference system time, which attempts to allow results to transfer to different experimental hardware. Ultimately the paper finds that none of the chosen methods give large improvements over baselines when controlling for compute budget, and the reviewers agree that this is a useful negative result. The paper received borderline scores. The major concern raised in the original reviews, which I agree with, is that the experiments used techniques that are not widely used, which limits the impact of the findings. However, the paper has significantly expanded this in the revision, for example comparing with well publicised recent work like the Sophia optimizer. While the reviewers agree that the revision has significantly improved the paper, some chose to focus their scores on the original submission. Overall, while I am not surprised by the negative results, I think they are a useful reminded to the community to focus on fair comparison, and I recommend acceptance.